

# Revision of the Tanzanian dicynodont *Dicynodon huenei* (Therapsida: Anomodontia) from the Permian Usili Formation

Christian F. Kammerer

North Carolina Museum of Natural Sciences, Raleigh, NC, USA

## ABSTRACT

A single species of the dicynodontoid dicynodont *Dicynodon* is currently recognized from the late Permian Usili Formation of Tanzania: *Dicynodon huenei* Haughton, 1932. Restudy of the known Tanzanian materials of *D. huenei* demonstrates that they represent two distinct morphotypes, here considered separate taxa. The holotype of *D. huenei* is not referable to *Dicynodon* and instead is transferred to the genus *Daptocephalus* (but retained as a valid species, *Daptocephalus huenei* comb. nov.). A number of published dicynodontoid specimens from the Usili Formation, however, are referable to *Dicynodon*, and are here recognized as a new species (*Dicynodon angielczyki* sp. nov.) *Dicynodon angielczyki* can be distinguished from its South African congener *Dicynodon lacerticeps* by the presence of an expansion of the squamosal and jugal beneath the postorbital bar and a curved, posterolateral expansion of the squamosal behind the temporal fenestra. Inclusion of *Dicynodon angielczyki* and *D. huenei* in a phylogenetic analysis supports their referral to *Dicynodon* and *Daptocephalus* (respectively). These results indicate higher basinal endemism in large late Permian dicynodonts than previously thought, a sharp contrast to the cosmopolitanism in the group in the earliest Triassic.

## INTRODUCTION

The Usili Formation is a sedimentary unit of late Permian age exposed in the Ruhuhu Basin at the southwestern edge of Tanzania (*Wopfner, 2002*; *Sidor & Nesbitt, 2018*). The Ruhuhu Basin has been recognized as fossiliferous since the initial geological surveys of Stockley (*Stockley, 1932*; *Stockley & Oates, 1931*), who collected a number of therapsid fossils in what is now considered the Usili Formation and sent them to the South African Museum (Cape Town, South Africa) for study. *Haughton (1932)* initially described these fossils, and named four new dicynodont species based on Stockley's collections: *Dicynodon huenei*, *Dicynodon tealei*, *Megacyclops rugosus*, and *Pachytegos stockleyi*. Of these taxa, only *D. huenei* is considered valid today: *Dicynodon tealei* and *M. rugosus* are considered rhachiocephalids of dubious validity (*Brink, 1986*; *Kammerer, Angielczyk & Fröbisch, 2011*) and *P. stockleyi* is considered a probable synonym of *Endothiodon bathystoma*, a taxon better known from the Karoo Basin of South Africa (*Cox & Angielczyk, 2015*).

Corresponding author
Christian F. Kammerer,
christian.kammerer@
naturalsciences.org

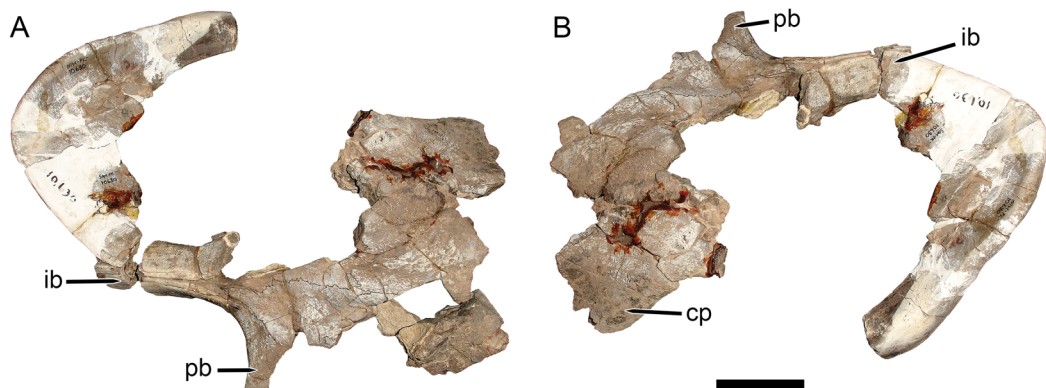

**Figure 1 Main cranial fragments of SAM-PK-10630, holotype of *Dicynodon huenei* (=*Daptocephalus huenei* comb. nov.).** Specimen in (A) dorsal and (B) left semi-lateral views. Note narrow intertemporal bar made up of vertically-oriented postorbital bones. Abbreviations: cp, caniniform process; ib, intertemporal bar; pb, postorbital bar. Scale bar equals five cm. Photos: Christian Kammerer.

*Haughton (1932)* established *Dicynodon huenei* based on a single specimen: SAM-PK-10630, a fragmentary partial skull (Fig. 1) and some associated postcranial elements, most notably the majority of a left scapula, from Stockley's locality B2. He did not explicitly differentiate the new taxon from the many existing species of *Dicynodon*, and the diagnostic features listed for his new species (e.g., short, wide snout, interorbital width greater than intertemporal) are present in many other dicynodontoids. Additional specimens of Usili Formation dicynodontoids were later collected by Ernst Nowack and sent to Tübingen, Germany for study by *von Huene (1942)* (*Nowack, 1937*). *von Huene (1942)* named the new species *Dicynodon bathyrhynchus* (currently *Euptychognathus bathyrhynchus*; *Kammerer, Angielczyk & Fröbisch, 2011*) for one of these specimens (GPIT/RE/7104), but referred the majority of Nowack's skulls to *D. huenei*, albeit transferring this species to the genus *Platypodosaurus*. *Platypodosaurus* is a problematic taxon originally established for large dicynodont postcranial remains from South Africa (*Owen, 1880*), and subsequent authors have not accepted Huene's referral of *D. huenei* to the genus, instead retaining it as a species of *Dicynodon* (*Haughton & Brink, 1954*; *King, 1988*). *Kammerer, Angielczyk & Fröbisch (2011)*, in their comprehensive global revision of *Dicynodon*, recognized *D. huenei* as a valid species and considered all dicynodontoid specimens from the Usili Formation (with the exception of the aforementioned *Euptychognathus bathyrhynchus*) to be referable to this taxon. This included Tanzanian specimens previously referred to the typically South African species *Dicynodon lacerticeps* (*Wild et al., 1993*) and *Daptocephalus leoniceps* (*Gay & Cruickshank, 1999*). *Kammerer, Angielczyk & Fröbisch (2011)* also referred a number of dicynodontoid specimens from the upper Madumabisa Mudstone Formation (Luangwa Basin, Zambia) to *D. huenei*, notably NHMUK PV R37005 (formerly TSK 14), a nearly complete skeleton described by *King (1981)* as a specimen of *Dicynodon trigonocephalus*. Finally, *Angielczyk et al. (2014)* provided further details on the distribution and anatomy of the Zambian specimens, and discussed the rationale behind their referral to *D. huenei*.

Recent expeditions (since 2007) to the Ruhuhu and Luangwa basins, led by researchers from the University of Washington and Field Museum of Natural History (USA), have collected a wealth of new therapsid fossils from the Usili and Madumabisa Mudstone Formations (*Angielczyk et al., 2009*, *2014*; *Weide et al., 2009*; *Sidor et al., 2010*; *Angielczyk & Cox, 2015*; *Huttenlocker, Sidor & Angielczyk, 2015*; *Huttenlocker & Sidor, 2016*; *Sidor & Nesbitt, 2018*). Among these specimens are numerous Zambian dicynodontoids, including well-preserved skulls matching NHMUK PV R37005 in general morphology but showing clear differences from all Usili Formation *D. huenei* specimens. These specimens are currently being described (K Angielczyk, 2019, personal communication), so will not be discussed in depth here, but suggest that there is at least species-level distinction between Zambian and Tanzanian "*D. huenei*", contra *Kammerer, Angielczyk & Fröbisch (2011)* and *Angielczyk et al. (2014)*.

Re-examination of the fragmentary holotype of *D. huenei*, SAM-PK-10630, provides additional evidence that *Kammerer, Angielczyk & Fröbisch (2011)* were overly conservative in treating all *Dicynodon* materials from Tanzania and Zambia as a single species. Because of the incompleteness of the holotype, the *Kammerer, Angielczyk & Fröbisch (2011)* diagnosis of *D. huenei* was based primarily on the series of complete Usili Formation dicynodontoid skulls housed in collections in Cambridge (UK) and Tübingen, notably UMZC T1089, GPIT/RE/7175, and GPIT/RE/7177. However, restudy of SAM-PK-10630 shows that, although damaged, this specimen differs in several important regards from those better known skulls. Here, I present a critical review of all published "*D. huenei*" material from Tanzania, argue that two distinct species are represented in this assemblage, and discuss the phylogenetic and biogeographic implications of this conclusion.

### Nomenclatural acts

The electronic version of this article in portable document format will represent a published work according to the International Commission on Zoological Nomenclature (ICZN), and hence the new names contained in the electronic version are effectively published under that Code from the electronic edition alone. This published work and the nomenclatural acts it contains have been registered in ZooBank, the online registration system for the ICZN. The ZooBank Life Science Identifiers (LSIDs) can be resolved and the associated information viewed through any standard web browser by appending the LSID to the prefix http://zoobank.org/. The LSID for this publication is: urn:lsid:zoobank.org:pub:714AFA18-EB4D-4A35-B948-B4E7CB046A77. The online version of this work is archived and available from the following digital repositories: PeerJ, PubMed Central, and CLOCKSS.

## COMPARATIVE BACKGROUND

*Kammerer, Angielczyk & Fröbisch (2011)* reviewed the original dicynodont genus, *Dicynodon Owen, 1845*, which had become a notorious wastebasket taxon. As part of this revision, they resurrected the genus *Daptocephalus* (long considered synonymous with *Dicynodon*; see *Cluver & King, 1983*; *King, 1988*) for the large South African species *Daptocephalus leoniceps*, recognizing it as distinct from the type species

*Dicynodon lacerticeps* based on morphometric analysis and discrete-state characters. In their phylogenetic analyses, *Kammerer, Angielczyk & Fröbisch (2011)* never recovered *Daptocephalus leoniceps* and *Dicynodon lacerticeps* as sister-taxa, supporting recognition of a separate genus for the former. More recent studies have maintained the distinction between *Dicynodon lacerticeps* and *Daptocephalus leoniceps* (*Botha-Brink, Huttenlocker & Modesto, 2014*; *Jasinoski et al., 2014*; *Angielczyk & Kammerer, 2017*; *Olivier et al., 2019*), and detailed analysis of the stratigraphic distributions of these two taxa indicates that they had somewhat different ranges (*Viglietti et al., 2016*; *Viglietti, Smith & Rubidge, 2018*).

*Kammerer, Angielczyk & Fröbisch (2011)* considered *Dicynodon lacerticeps* and *Daptocephalus leoniceps* to be restricted to the Karoo Basin of South Africa, and considered extra-basinal Permian dicynodontoid records to mostly represent distinct, locally-endemic taxa (e.g., *Jimusaria* and *Turfanodon* in China; *Peramodon* and *Vivaxosaurus* in Russia; *Gordonia* in Scotland). They recognized only two Permian dicynodontoid taxa with international ranges: *Euptychognathus bathyrhynchus* (recorded in South Africa and Tanzania) and *Dicynodon huenei* (recorded in Tanzania and Zambia). *Euptychognathus bathyrhynchus* is a rare taxon, and of the four recorded specimens, only one (the holotype) was found in Tanzania. By contrast, *Dicynodon huenei* sensu *Kammerer, Angielczyk & Fröbisch (2011)* is relatively common: they referred 14 specimens to the species, and *Angielczyk et al. (2014)* referred additional specimens from Zambia to *D. huenei*, with further unprepared or juvenile skulls considered possibly referable.

Both *Kammerer, Angielczyk & Fröbisch (2011)* and *Angielczyk et al. (2014)* used the large, complete skull UMZC T1089 as their primary exemplar of *D. huenei*, rather than the fragmentary and poorly-preserved holotype (SAM-PK-10630), and referral of additional specimens to the species was largely based on comparisons with the former skull. They considered the main diagnostic feature of the species to be an expanded section of the zygoma beneath the postorbital bar, thickened to form a large plate at the posteroventral edge of the orbit and making the zygoma appear bowed outward in dorsal or ventral view, as is very evident in UMZC T1089 and the similar specimens GPIT/RE/7175 (Figs. 2A and 2C) and GPIT/RE/7177 (Fig. 3A). *Kammerer, Angielczyk & Fröbisch (2011)* stated that, although incomplete, this morphology was present in SAM-PK-10630, and *Angielczyk et al. (2014)* noted its presence in NHMUK PV R37005 and other Zambian specimens.

It is true that the zygomatic arches of SAM-PK-10630 (based on the description of *Haughton (1932)* and the existing fragments) and the Zambian specimens discussed by *Angielczyk et al. (2014)* are more bowed than those of the South African *Dicynodon lacerticeps* (Fig. 4A) when viewed dorsally. As mentioned above, however, new Zambian dicynodontoid specimens currently under study show several distinctive features differentiating them from UMZC T1089 and similar specimens, including a septomaxillary-lacrimal contact and presence of a deep occipital excavation above the paroccipital process (C Kammerer, 2019, personal observations). Furthermore, although the zygoma in the Zambian specimens is indeed bowed, they do not show the same style of zygomatic expansion as in the Tanzanian material. In GPIT/RE/7175, GPIT/RE/7177, and UMZC

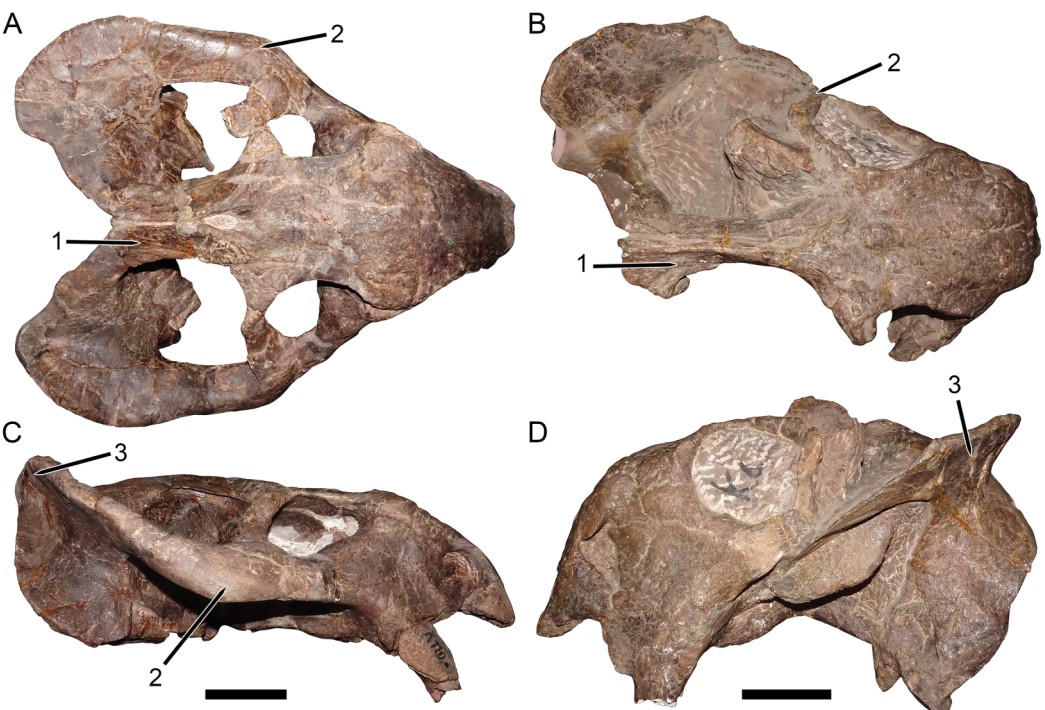

**Figure 2 Comparisons between the two morphotypes of Usili Formation dicynodontoids previously included in "Dicynodon huenei".** GPIT/RE/7175, referred specimen of *Dicynodon angielczyki* sp. nov., in (A) dorsal and (C) right lateral views. GPIT/RE/9316, referred specimen of *Daptocephalus huenei*, in (B) dorsal and (D) left lateral views. Distinguishing features of the two morphotypes labeled on the figure: (1) width of the intertemporal bar and orientation of the postorbitals (relatively broad with horizontal postorbitals in *Dicynodon angielczyki* vs. relatively narrow with vertical postorbitals in *Daptocephalus huenei*), (2) thickness of the zygoma below the postorbital bar (squamosal and jugal expanded in height, with distinct lateral bowing visible in dorsal view, in *Dicynodon angielczyki* vs. narrow base with no bowing in *Daptocephalus huenei*), (3) angulation of squamosal at junction between zygomatic and quadrate rami (highly acute with relatively narrow flange at posterior end of zygomatic ramus in *Dicynodon angielczyki* vs. less acute with broad flange in *Daptocephalus huenei*). Note also the generally taller skull and especially deeper snout in *Daptocephalus huenei*, as well as the proportionally broader interorbital region and longer intemporal region of that species. The features noted above are generally typical of *Dicynodon* and *Daptocephalus* at the generic level (*Kammerer, Angielczyk & Fröbisch, 2011*), with the exception of the squamosal expansion, which is a species level autapomorphy of *Dicynodon angielczyki*. Scale bars equal five cm. Photos: Christian Kammerer.                                      

T1089, the anterior process of the squamosal is enlarged, forming a thickened edge to the zygoma postorbitally. However, the jugal in this region is also expanded dorsoventrally, separating the postorbital from the squamosal. In the Zambian specimens, as in most dicynodontoids, there is no such jugal expansion and the postorbital still contacts the dorsal edge of the squamosal. Further commentary on the relationships of the Zambian specimens will have to await their full description, but at present they should not be considered conspecific with the Tanzanian material.

Only part of the zygoma is preserved in SAM-PK-10630. More extensively preserved, however, are the skull roof (including the intertemporal bar) and right snout (Fig. 1). Recent re-examination of these fragments demonstrates that they differ in several important ways from the "standard *D. huenei*" morphotype represented by GPIT/RE/7175,

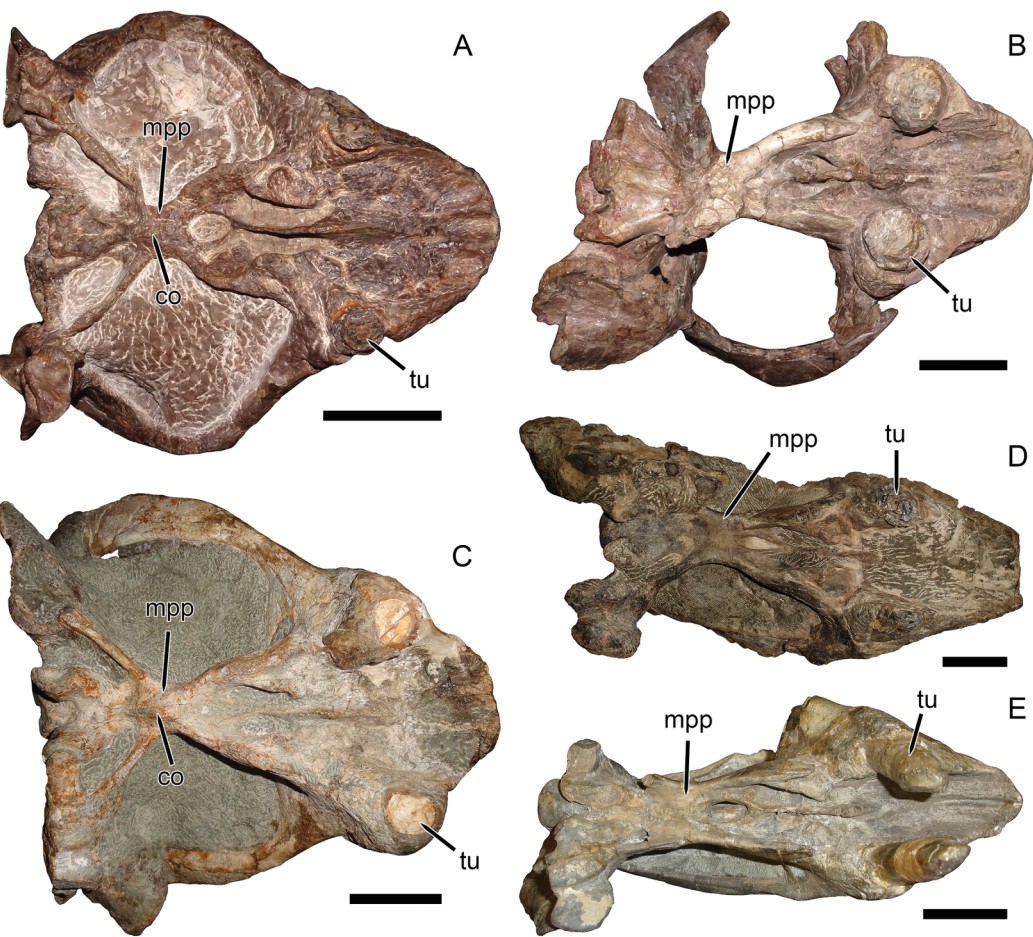

**Figure 3 Palatal comparisons between *Dicynodon* and *Daptocephalus*.** (A) GPIT/RE/7177, referred specimen of *Dicynodon angielczyki* sp. nov. (B) GPIT/RE/9641, referred specimen of *Daptocephalus huenei*. (C) RC 38, referred specimen of *Dicynodon lacerticeps* (holotype of *D. aetorhamphus*). (D) NHMUK PV OR 47047, holotype of *Daptocephalus leoniceps*. (E) BP/1/2784, referred specimen of *Daptocephalus leoniceps*. All specimens in ventral view, with anterior right. GPIT/RE/9641 is badly anteroposteriorly distorted, whereas NHMUK PV OR 47047 and BP/1/2784 are slightly laterally compressed and GPIT/RE/7177 is slightly dorsoventrally compressed (RC 38 is largely undistorted). However, note narrower span between tusks in *Daptocephalus* specimens regardless of style of deformation, and their narrower pre-caniniform region of the premaxilla relative to *Dicynodon*. Note also the relatively narrow, distinctly constricted median pterygoid plate bearing a sharp median ridge (the crista oesophagea) in *Dicynodon*. In *Daptocephalus*, the median pterygoid plate is comparatively broad, less sharply constricted from the anterior and posterior pterygoid rami, and lacks a distinct crista oesophagea. Abbreviations: co, crista oesophagea; mpp, median pterygoid plate; tu, tusk. Scale bars equal five cm. Photos: Christian Kammerer.

GPIT/RE/7177, and UMZC T1089. In SAM-PK-10630, the intertemporal exposures of the postorbitals are nearly vertical, whereas in the "standard *D. huenei*" morphotype they are more horizontal. A median interorbital ridge is present on the skull roof in SAM-PK-10630, but absent in "standard *D. huenei*". The snout of SAM-PK-10630 is relatively tall and sharply-sloping, but relatively low and gradually-sloping in "standard *D. huenei*". Tusks are ventrally directed in SAM-PK-10630, but anteroventrally angled in "standard *D. huenei*". Intriguingly, this set of characters is not unique to SAM-PK-10630 among Usili

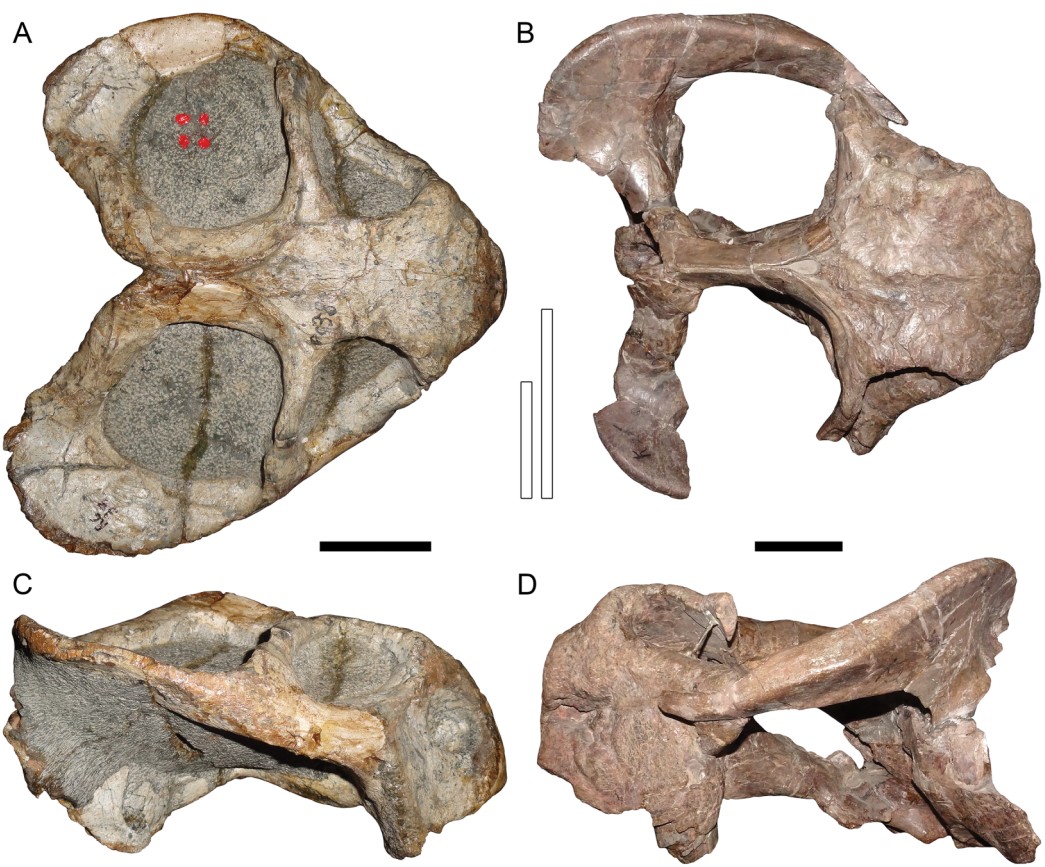

**Figure 4 Comparisons between similarly-distorted skulls of *Dicynodon* and *Daptocephalus*.** RC 38, a referred specimen of *Dicynodon lacerticeps* (holotype of *D. trigonocephalus*), in (A) dorsal and (C) right lateral views. GPIT/RE/9641, a referred specimen of *Daptocephalus huenei*, in (B) dorsal and (D) left lateral views. Specimens scaled to equal anteroposterior length. White vertical bars between (A) and (B) illustrate the least interorbital width of RC 38 (left) and GPIT/RE/9641 (right), showing the greater interorbital width of *Daptocephalus* relative to *Dicynodon* even when their skulls are otherwise (and atypically) similar in shape due to anteroposterior compression. Note also the greater depth of the snout, more vertically-oriented postorbital contributions to the intertemporal bar, and broader, less sharply angled junction between the zygomatic and quadrate rami of the squamosal in *Daptocephalus*. Scale bars (black horizontal) equal five cm. Photos: Christian Kammerer.

Formation dicynodontoids, but is also evident in the more complete skulls GPIT/RE/9316 (Figs. 2B and 2D) and GPIT/RE/9641 (Figs. 3B, 4B and 4D). These specimens show additional differences separating them from the "standard *D. huenei*" morphotype, notably a taller, transversely narrower occiput (Fig. 5), proportionally broader interorbital region, proportionally longer, narrower intertemporal bar, and a less constricted median pterygoid plate lacking a distinct crista oesophagea. These other specimens also appear to lack the zygomatic expansion of "standard *D. huenei*," but have a more vertically-oriented, broadly rounded subtemporal arch. Comparison with SAM-PK-10630 indicates that the apparent expansion of the subtemporal arch in that specimen is just the result of displacement of the more vertical, posterior section of the bar seen in specimens like GPIT/RE/9641, rather than actual dorsoventral and transverse expansion as in UMZC T1089.

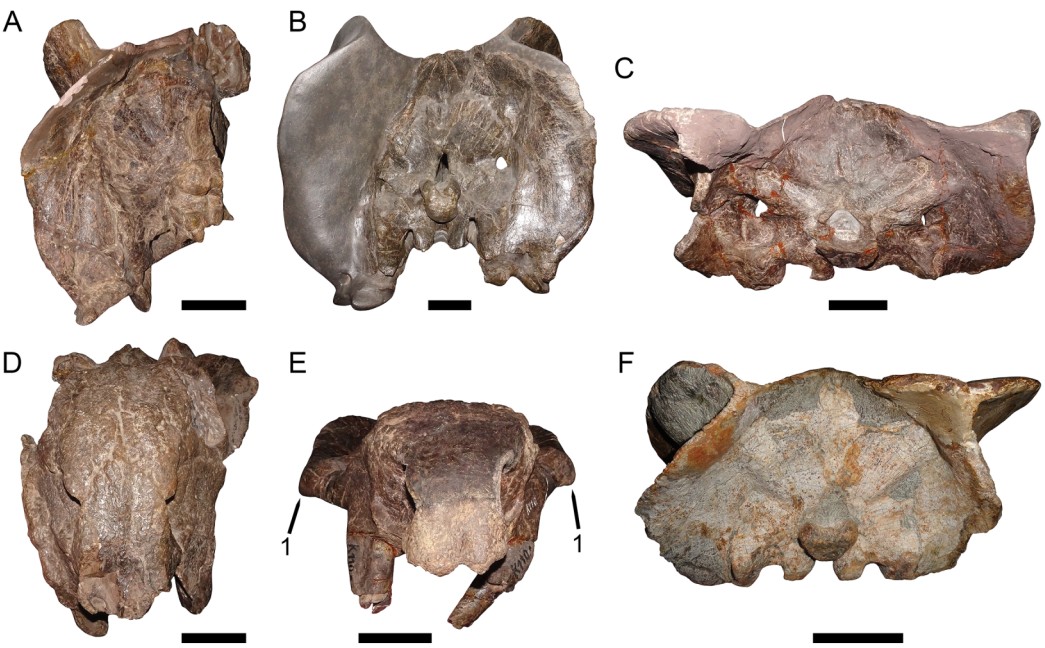

**Figure 5 Occipital and anterior comparisons between *Dicynodon* and *Daptocephalus*.** GPIT/RE/9316, referred specimen of *Daptocephalus huenei*, in (A) occipital and (D) anterior views. MB.R.992, referred specimen of *Daptocephalus leoniceps*, in (B) occipital view. GPIT/RE/7175, referred specimen of *Dicynodon angielczyki* sp. nov., in (C) occipital and (E) anterior views. RC 23, referred specimen of *Dicynodon lacerticeps* (holotype of *D. cadlei*), in (F) occipital view. Note the broader, lower occiput and snout of *Dicynodon* specimens, and (1) the autapomorphic lateral expansions of the zygomatic arch in *Dicynodon angielczyki*. Scale bars equal five cm. Photos: Christian Kammerer.

The features separating SAM-PK-10630, GPIT/RE/9316, and GPIT/RE/9641 from the "standard *D. huenei*" morphotype include many of the characters cited by *Kammerer, Angielczyk & Fröbisch (2011)* as differentiating *Daptocephalus* from *Dicynodon*. The relatively tall, sharply-sloping snout, ventrally-oriented tusks, vertically-oriented, broadly rounded subtemporal arches, narrow, extremely elongate intertemporal bar, and vertical orientation of the postorbitals in the intertemporal bar are all characteristic features of the genus *Daptocephalus*. The presence of an interorbital ridge, a relatively broad interorbital region, and a weakly constricted median pterygoid plate lacking a crista oesophagea also differentiate *Daptocephalus* from *Dicynodon* (C Kammerer, 2019, personal observations), and are newly recognized here as diagnostic features of the former genus (Figs. 3 and 4; also see Table 1). (It is worth noting here that the smallest specimens otherwise identifiable as *Dicynodon lacerticeps* also lack a crista oesophagea, so the presence of this character may be ontogenetically variable in that species. However, all known specimens of *Daptocephalus leoniceps*, most of which are very large skulls, lack a crista oesophagea, so in mature specimens of the two genera this appears to be a reliable differentiator.) In total, the suite of features listed above indicates that two morphotypes of dicynodontoid are present in the Usili Formation (in addition to the singleton record of *Euptychognathus*, which can readily be distinguished from both morphotypes by its extremely tall snout and U-shaped naso-frontal ridge). These morphotypes are most

**Table 1 Cranial measurements (in cm) of the more complete specimens of *Daptocephalus huenei* and *Dicynodon angielczyki*.**

| | *Daptocephalus huenei* | | | *Dicynodon angielczyki* | | |
|---|---|---|---|---|---|---|
| | GPIT/RE/9316 | GPIT/RE/9317 | GPIT/RE/9641 | GPIT/RE/7175 | GPIT/RE/7177 | UMZC T1089 |
| Dorsal skull length | 23.6 | ~25 | 23.0 | 27.1 | 21.1 | 27.3 |
| Basal skull length | 28.2 | ~28 | 29.8 | 29.2 | 22.2 | 31.5 |
| Snout length | 9.4 | NA | 5.1 | 11.3 | 6.3 | 10.2 |
| Interorbital width (minimum) | 8.1 | 8.5 | 10.2 | 7.8 | 5.5 | 7.9 |
| Anterior intertemporal width | 5.6 | 5.0 | 6.7 | 6.9 | 5.7 | 8.0 |
| Posterior intertemporal width | 3.1 | 3.0 | 3.0 | 5.8 | 3.5 | 6.8 |
| Temporal fenestra length (left maximum) | 17.9 | NA | 17.0 | 16.7 | 13.7 | 19.4 |
| Temporal fenestra length (right maximum) | NA | NA | 16.8 | 17.2 | NA | 17.3 |
| Median pterygoid plate width (minimum) | 3.7 | 4.2 | 4.7 | 3.1 | 2.3 | 2.9 |

**Note:**

Anterior intertemporal width taken at the junction between the intertemporal and postorbital bars, posterior intertemporal width taken at the junction between the intertemporal bar and occiput, following *Kammerer, Angielczyk & Fröbisch (2011)*.

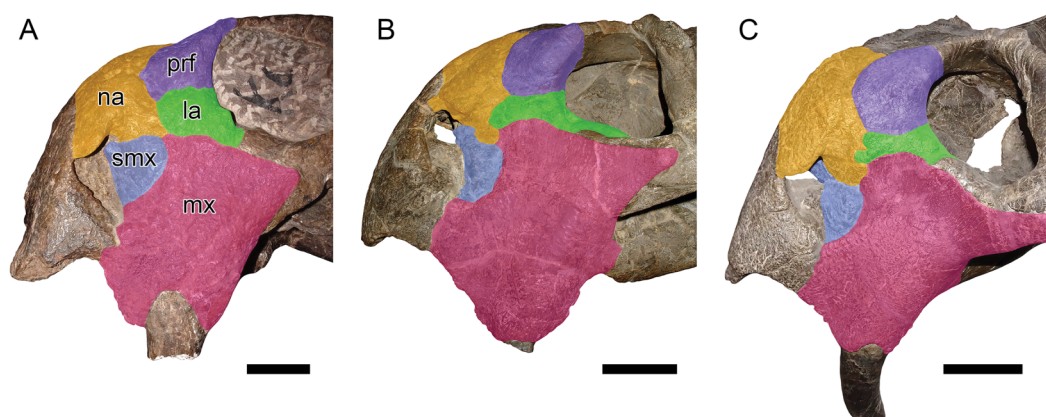

**Figure 6 Comparisons between *Daptocephalus huenei* comb. nov. and *Daptocephalus leoniceps*.** (A) GPIT/RE/9316, referred specimen of *Daptocephalus huenei*, in left lateral view. (B) UCMP 33431, referred specimen of *Daptocephalus leoniceps*, in right lateral view (mirrored for comparative purposes). (C) MB.R.992, referred specimen of *Daptocephalus leoniceps*, in left lateral view. Major facial bones colored to show arrangement in various specimens and highlight comparatively large size of the lacrimal in *Daptocephalus huenei*, here interpreted as autapomorphic for the species. Abbreviations: la, lacrimal; mx, maxilla; na, nasal; prf, prefrontal; smx, septomaxilla. Scale bars equal five cm. Photos: Christian Kammerer.

similar, among known dicynodonts, to the South African taxa *Dicynodon lacerticeps* (for the so-called "standard *D. huenei*" morphotype) and *Daptocephalus leoniceps* (for the *D. huenei* holotype and similar specimens) and are here considered congeneric with them (see Phylogenetic Analysis for further rationale behind this). However, each varies somewhat from their Karoo counterparts. The holotype SAM-PK-10630 and GPIT/RE/9316 show dorsoventrally taller lacrimals than in *Daptocephalus leoniceps* (Fig. 6; see further discussion below), although otherwise Tanzanian and South African *Daptocephalus*

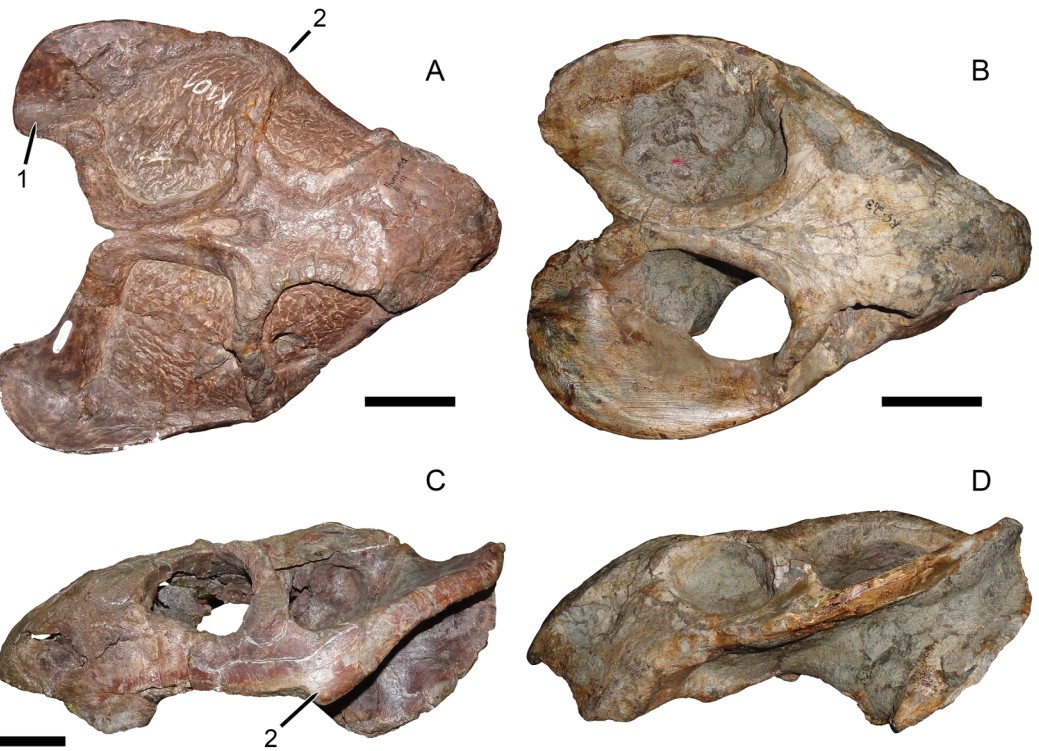

**Figure 7 Comparisons between *Dicynodon angielczyki* sp. nov. and *Dicynodon lacerticeps*.** GPIT/RE/ 7177, referred specimen of *Dicynodon angielczyki*, in (A) dorsal view. RC 23, referred specimen of *Dicynodon lacerticeps* (holotype of *D. cadlei*), in (B) dorsal and (D) left lateral views. UMZC T1089, holotype of *Dicynodon angielczyki*, in (C) left lateral view. Autapomorphies of *Dicynodon angielczyki* labeled on the figure: (1) medially-curved squamosal flange at posterolateral edge of temporal fenestra (extensively restored in plaster in this specimen, but sharp curvature at rear edge of squamosal shows it was present; this morphology is clearly preserved in other *Dicynodon angielczyki* specimens, see Figs. 2A and 7A); (2) zygomatic arched bowed laterally (shown in A) and dorsoventrally expanded (shown in C) below and immediately behind the postorbital bar. Scale bars equal five cm. Photos: Christian Kammerer.

specimens are very similar. GPIT/RE/7175, GPIT/RE/7177, UMZC T1089 and other representatives of the "standard *D. huenei*" morphotype are more easily distinguished from their Karoo counterpart *Dicynodon lacerticeps* by their massively expanded suborbital zygoma and bowed subtemporal arches (Fig. 7).

Here, the two morphotypes of Usili Formation dicynodontoid formerly included within *Dicynodon huenei* are recognized as distinct species, which are closely related to (but distinct from) the well-known South African dicynodontoid taxa *Dicynodon lacerticeps* and *Daptocephalus leoniceps*. Confoundingly, the holotype of *Dicynodon huenei* is not part of the "standard *D. huenei*" morphotype that has been the basis for most of the recent literature on the species, but instead is referable to the genus *Daptocephalus* (as *Daptocephalus huenei* comb. nov.) No preexisting species names are available for what was previously considered "standard *D. huenei*," requiring the establishment of a new species for this morphotype.

Almost all known dicynodontoid specimens from the Usili Formation can be referred to one of the two taxa recognized here (see full list of referred materials in the

Systematic Paleontology section below). However, a few published specimens from the Usili Formation cannot be confidently identified to species at present. *Wild et al. (1993)* described a dicynodont skull from the Usili Formation that they referred to the South African species *Dicynodon lacerticeps*. The whereabouts of this specimen are currently unknown; no specimen number or institutional depository were listed in its description. Based on their description, it does appear to be a dicynodontoid, and their Fig. 4C shows the skull as having vertically-oriented postorbitals in the intertemporal bar. As such, it may represent a specimen of *Daptocephalus huenei*. Until this specimen is relocated and can be examined in detail, however, this record should only be considered Dicynodontoidea indet. UMZC T1280, referred to *Dicynodon huenei* by *Kammerer, Angielczyk & Fröbisch (2011)*, consists of a partial mandible from Stockley's Locality B4/7. This jaw has a deeper symphysis than UMZC T1123, suggesting that it could be *Daptocephalus huenei* (the jaw symphysis is relatively deep in *Daptocephalus leoniceps*), but until associated jaws with definite skulls are known for *Daptocephalus huenei*, this specimen should also be regarded as Dicynodontoidea indet.

## SYSTEMATIC PALEONTOLOGY

**Synapsida** *Osborn, 1903*
**Therapsida** *Broom, 1905*
**Anomodontia** *Owen, 1860a*
**Dicynodontia** *Owen, 1860a*
**Dicynodontoidea** *Olson, 1944*

**Definition:** All taxa more closely related to *Dicynodon lacerticeps* Owen, 1845 than *Oudenodon bainii* *Owen, 1860b* or *Emydops arctatus* (*Owen, 1876*) (*Kammerer & Angielczyk, 2009*).

**Remarks:** *Kammerer & Angielczyk (2009)* followed *Cluver & King (1983)* in treating Dicynodontoidea as a superfamily, and assigned authorship of it to *Cope (1871)*. *Cope (1871)* was the first to use the orthography Dicynodontidae for the family containing *Dicynodon*, so he was considered to be author of Dicynodontoidea by the Principle of Coordination (Art. 36.1 of The Code; *ICZN, 1999*). Further research has revealed two problems with this, however. First, although his usage predated standardized familial suffixes in zoological nomenclature, *Owen's (1860a)* explicit treatment of his taxon Dicynodontia as a family is considered equivalent to the establishment of Dicynodontidae under Art. 11.7.1.3 of The Code (*ICZN, 1999*). Thus, *Owen (1860a)* is the author of both the unranked higher taxon Dicynodontia and the family Dicynodontidae. Second, the earliest usage of the taxon Dicynodontoidea in the literature was not as a superfamily, but explicitly as an infraorder by *Olson (1944)*. Thus, Dicynodontoidea represents a higher taxon outside the strictures of The Code and is not coordinate with Dicynodontidae (despite having the standard suffix for superfamily; this is the case for numerous higher taxa, for example, Asteroidea, Hyracoidea, etc.).

*Daptocephalus* van Hoepen, 1934

**Type species:** *Daptocephalus leoniceps* (Owen, 1876).

**Included species:** *Daptocephalus huenei* (Haughton, 1932).

**Diagnosis:** Dicynodontoid characterized by the combination of a proportionally tall, steeply sloping snout, ventrally-directed tusks, narrow palatal portion of the premaxilla, median interorbital ridge, long, extremely narrow intertemporal bar, vertical orientation of the postorbitals in the intertemporal bar, zygomatic ramus of squamosal tall and broadly rounded at posterior end, and broad median pterygoid plate lacking a well-developed crista oesophagea. Premaxillary beak tip not sharply "hooked" as in *Dinanomodon*. Weak anterior processes of frontals present, but not elongate, attenuate processes nearing or contacting premaxilla as in *Dinanomodon* and *Vivaxosaurus*.

*Daptocephalus huenei* (Haughton, 1932) comb. nov.

(Figs. 1, 2B, 2D, 3B, 4B, 4D, 5A, 5D and 6A)

**Holotype:** SAM-PK-10630, a fragmentary skull, left scapula, and fragments of postcranial elements from Stockley's locality B2, Kiwohe, Ruhuhu Basin, Tanzania.

**Referred material:** All referred material is from the Ruhuhu Basin of southwestern Tanzania. GPIT/RE/9316 (=K6), a partial skull (missing the right temporal arch and half of occiput) from Kingori; GPIT/RE/9317 (=K6), a partial skull (missing the snout tip, temporal arches, and edges of the occiput) from Kingori; GPIT/RE/9641 (=K2), a badly distorted skull from Kingori; SAM-PK-10634, a partial skull consisting of the snout and fragmentary skull roof from Stockley's locality B16; UMZC T799, a skull roof and associated tusk from Stockley's locality B4/6; UMZC T983, a largely unprepared partial skull (missing the temporal arches) from Stockley's locality B4/4; UMZC T1282, a small partial skull roof broken into sections (the anteriormost of which appears too large to fit on the rest of the skull and likely represents a different individual) from an uncertain Usili locality.

**Diagnosis:** Distinguished from *Daptocephalus leoniceps* by its proportionally taller lacrimal and possibly by a longer premaxillary beak tip.

**Remarks:** The rationale for taxonomically differentiating these specimens from *Dicynodon* and referring them to *Daptocephalus* is provided above. Known material of *Daptocephalus huenei* is extremely similar to that of the South African type species *Daptocephalus leoniceps*. However, there is at least one consistent difference: Tanzanian *Daptocephalus* specimens have an enlarged lacrimal compared to their South African congeners, with a taller facial portion between the prefrontal and maxilla (Fig. 6). A relatively tall lacrimal is present in the holotype, SAM-PK-10630, meaning that despite the extreme inadequacy of that specimen, it is diagnosable, and the name *D. huenei* can be retained for this morphotype. Although less certain of an autapomorphic feature due to frequent damage

(it is missing in the holotype), in specimens of *Daptocephalus huenei* that preserve the tip of the premaxillary beak, this structure is proportionally longer than in comparably intact, well-preserved *Daptocephalus leoniceps* specimens. This is especially evident in GPIT/RE/9316 (Figs. 2D and 6A).

Most of the published material referable to *Daptocephalus huenei* is rather poor, and attempting a thorough redescription of this species here would be premature. The best-preserved of the described specimens, GPIT/RE/9316 (Figs. 2B, 2D, 5A and 5D), is missing most of the right side of the skull and its braincase is largely unprepared; other specimens are either highly distorted, fragmentary, or unprepared. However, recent expeditions to the Ruhuhu Basin have recovered several new specimens of *Daptocephalus huenei* (K Angielczyk, 2019, personal communication), including a complete, relatively well-preserved skull (NMT RB43; C Kammerer, 2019, personal observations, see also Supplementary Data 4 of *Angielczyk, Hancox & Nabavizadeh, 2018*). The description of these new specimens should greatly improve our understanding of the anatomy of *Daptocephalus huenei*, and may reveal additional features distinguishing it from *Daptocephalus leoniceps*.

**Dicynodon** Owen, 1845

**Type species:** *Dicynodon lacerticeps* Owen, 1845.

**Included species:** *Dicynodon angielczyki* sp. nov.

**Diagnosis:** Dicynodontoid characterized by the combination of a relatively low, weakly-sloping snout, anteroventrally-directed tusks, broad, usually squared-off palatal portion of the premaxilla, median pterygoid plate distinctly constricted relative to rest of pterygoid in adults, bearing well-developed crista oesophagea, horizontal orientation of the postorbitals in the intertemporal bar, relatively short intertemporal bar, and zygomatic and quadrate rami of the squamosal forming an acute angle.

**Dicynodon angielczyki** sp. nov.

(Figs. 2A, 2C, 3A, 5C, 5E, 7A, 7C and 8–12)

**Holotype:** UMZC T1089, a complete skull from Stockley's locality B19, Kingori, Ruhuhu Basin, Tanzania.

**Referred material:** All referred material is from the Ruhuhu Basin of southwestern Tanzania. GPIT/RE/7175 (=K110), a complete skull from Kingori; GPIT/RE/7177 (=K101), a complete skull from Kingori; UMZC T979, a fragmented but nearly complete skull (missing parts of the snout roof and postorbital bars) from Stockley's locality B4/6; UMZC T982, a fragmented but nearly complete skull (missing parts of the palate and skull roof) from Stockley's locality B4/7; UMZC T1123, a complete skull, lower jaws, and possible skeletal elements preserved in association with two gorgonopsians from Stockley's locality B4/3; UMZC T1126, a partial skull roof from Stockley's locality B4/5.

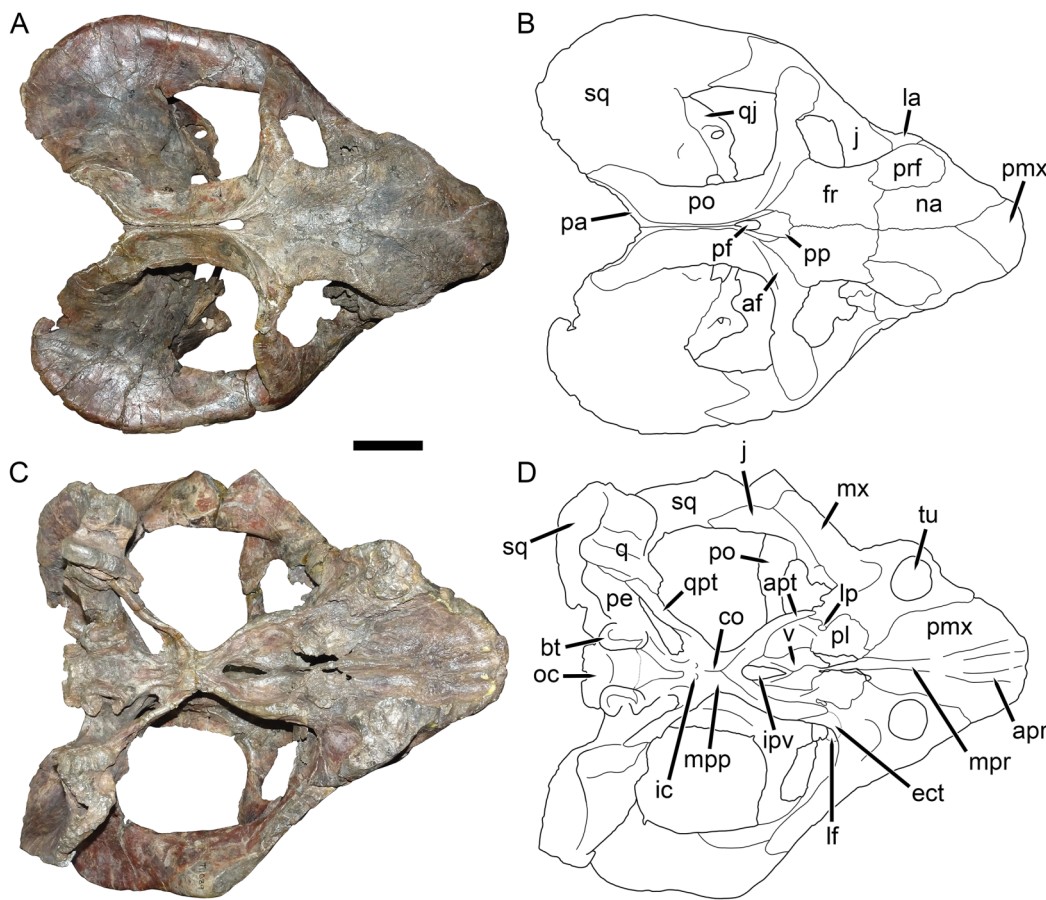

**Figure 8** UMZC T1089, holotype of *Dicynodon angielczyki* sp. nov. in dorsal and ventral views. (A) photograph and (B) interpretive drawing of the specimen in dorsal view. (C) photograph and (D) interpretive drawing of the specimen in ventral view. Abbreviations: af, adductor fossa of postorbital; apr, anterior palatal ridge; apt, anterior pterygoid ramus; bt, basal tuber; co, crista oesophagea; ect, ectopterygoid; fr, frontal; ic, internal carotid canal; ipv, interpterygoid vacuity; j, jugal; la, lacrimal; lf, labial fossa; lp, lateral palatal foramen; mx, maxilla; mpp, median pterygoid plate; mpr, median palatal ridge; na, nasal; oc, occipital condyle; pa, parietal; pe, periotic; pf, pineal foramen; pl, palatine; pmx, premaxilla; po, postorbital; pp, preparietal; prf, prefrontal; q, quadrate; qj, quadratojugal; qpt, quadrate ramus of pterygoid; sq, squamosal; tu, tusk; v, vomer. Scale bar equals five cm. Photos/drawings: Christian Kammerer.

**Diagnosis:** Distinguished from *Dicynodon lacerticeps* by expansion of the zygomatic ramus of the squamosal anteriorly, becoming dorsoventrally and mediolaterally swollen below the level of the postorbital bar; dorsoventral expansion of the jugal below the postorbital bar, such that the postorbital bone is widely separated from the squamosal; absence of a distinct postfrontal; and extension of the squamosal at the posterolateral corner of the temporal fenestra, forming a flange that curves slightly medially (Fig. 7).

**Etymology:** Named after the preeminent dicynodont researcher Kenneth Angielczyk, in particular recognition of his work on the Tanzanian and Zambian dicynodont faunas.

**Description:** Three intact, nearly-complete, thoroughly-prepared crania are known for *Dicynodon angielczyki*: GPIT/RE/7175 (Figs. 2A, 2C, 5C and 5E), GPIT/RE/7177

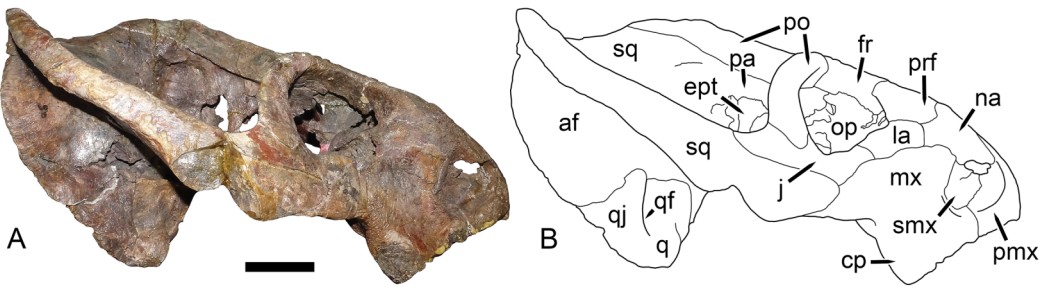

**Figure 9 UMZC T1089, holotype of *Dicynodon angielczyki* sp. nov. in right lateral view.** (A) photograph and (B) interpretive drawing. Abbreviations: af, adductor fossa of squamosal; cp, caniniform process of maxilla; ept, epipterygoid; fr, frontal; j, jugal; la, lacrimal; mx, maxilla; na, nasal; op, orbital plate; pa, parietal; pmx, premaxilla; po, postorbital; prf, prefrontal; q, quadrate; qf, quadratojugal foramen; qj, quadratojugal; smx, septomaxilla; sq, squamosal. Scale bar equals five cm. Photo/drawing: Christian Kammerer.

(Figs. 3A and 7A), and UMZC T1089 (Figs. 7C, 8–10). The following description is based primarily on UMZC T1089, which best illustrates the cranial sutures in this taxon (the skull roofs of the Tübingen specimens are somewhat overprepared). No associated mandibles are preserved with the aforementioned specimens, however, so the mandibular description is based on UMZC T1123. UMZC T1122–T1123 (Figs. 11 and 12) is a large block of jumbled fossils from Stockley's locality B4 (Katumbi viwili, also variously written as Katumbi vawili and Katumbi mwili), one of the most productive Permian fossil sites in the Ruhuhu Basin (*Gay & Cruickshank, 1999*; *Angielczyk, 2007*). The remains of two gorgonopsians and one dicynodont can be identified in this block. The two gorgonopsians appear to represent the same species, which can be recognized as a rubidgeine on the basis of the greatly expanded height of the zygomatic arch and extreme transverse width of the temporal region (*Kammerer, 2016*). Definite species level identification of these gorgonopsians is not possible at present due to incomplete exposure of their skulls (notably, the palates are not visible), but based on general proportions they are probably referable to the common Usili Formation rubidgeine *Sycosaurus nowaki*. The dicynodont is represented by a complete skull exposed in ventral view (33.0 cm basal skull length, making it slightly larger than the 31.5 cm holotype), some possible postcrania (although most of the postcranial material appears gorgonopsian), and a lower jaw disarticulated into its component rami. Only the posterior tip of the right mandibular ramus is exposed; the rest of the jaw descends into the block (Fig. 11B). However, the left mandibular ramus has been separated from the main block and fully prepared (Fig. 12). This specimen can be identified as a dicynodontoid on the basis of an enlarged labial fossa and the uniformly rugose palatine pad flush with the surrounding palate (as opposed to the condition in geikiids, where the palatine pad is smoother and flush with the rest of the palate anteriorly but raised and extremely rugose posteriorly), and can be recognized as *Dicynodon angielczyki* rather than *Daptocephalus huenei* based on the relatively narrow median pterygoid plate bearing a well-developed crista oesophagea. It can further be distinguished from *Euptychognathus bathyrhynchus* by the greater transverse width of the premaxilla and occiput and relatively low snout. Because of limited exposure of the skull,

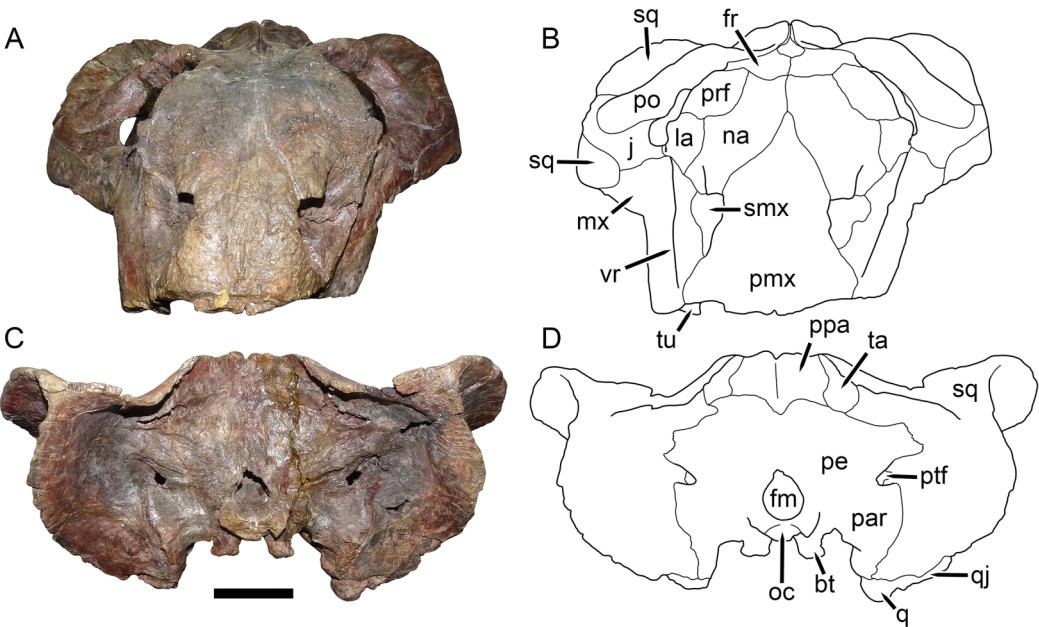

**Figure 10 UMZC T1089, holotype of *Dicynodon angielczyki* sp. nov. in anterior and posterior views.**
(A) photograph and (B) interpretive drawing of specimen in anterior view. (C) photograph and
(D) interpretive drawing of specimen in posterior view. Abbreviations: bt, basal tuber; fm, foramen
magnum; fr, frontal; j, jugal; la, lacrimal; mx, maxilla; na, nasal; oc, occipital condyle; par, paroccipital
process of periotic; pe, periotic; pmx, premaxilla; po, postorbital; ppa, postparietal; prf, prefrontal; ptf,
post-temporal fenestra; q, quadrate; qj, quadratojugal; smx, septomaxilla; sq, squamosal; ta, tabular; tu,
tusk; vr, vertical ridge on maxilla. Scale bar equals five cm. Photos/drawings: Christian Kammerer.

**Figure 11 UMZC T1122–T1123, a fossil block containing the jumbled remains of two gorgonopsians
and a specimen of *Dicynodon angielczyki* sp nov.** (A) entire block. (B) close-up of *Dicynodon
angielczyki* skull in ventral view, with gorgonopsian scapula removed and additional gorgonopsian
material lightened to highlight the dicynodont. Both jaw rami of this dicynodont specimen are preserved:
the right ramus is descending into the block and visible in (B), whereas the left ramus has been prepared
out and is shown in Fig. 12. Abbreviations: ar, articular; co, crista oesophagea; tu, tusk. Scale bars equal
five cm. Photos: Christian Kammerer.

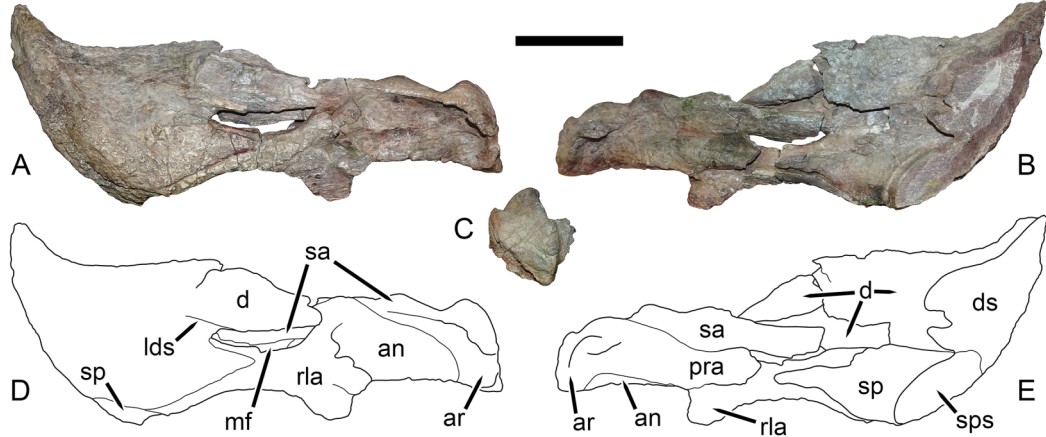

**Figure 12 Left mandibular ramus of UMZC T1123, referred specimen of *Dicynodon angielczyki* sp. nov.** Jaw in (A) left lateral, (B) right medial, and (C) posterior views with (D and E) interpretive drawings. Abbreviations: an, angular; ar, articular; d, dentary; ds, sagittal section through dentary; lds, lateral dentary shelf; mf, mandibular fenestra; pra, prearticular; rla, reflected lamina of angular; sa, surangular; sp, splenial; sps, sagittal section through splenial. Scale bar equals five cm. Photos/drawings: Christian Kammerer.

this specimen provides little data on the cranium of this species not already known in the specimens discussed above. However, it is important in providing the only available information on mandibular morphology for *Dicynodon angielczyki*, and is the basis for the mandibular description presented below.

The premaxilla of *Dicynodon angielczyki* is a fused median element forming a beak, as in all known dicynodonts. It makes a triangular contribution to the dorsal surface of the snout (Fig. 8), terminating above the nares (Fig. 9). A broad, low median ridge is present on the anterodorsal face of the premaxilla. Lateral to this ridge the premaxillary surface is noticeably sculptured (Fig. 10), probably associated with a keratinous beak covering in life. Ventrally, the premaxilla makes up a broad secondary palatal plate extending posteriorly to contact the palatines and vomer. Well-developed, parallel anterior palatal ridges are present at the tip of the beak. These do not converge posteriorly, but are confluent with a weak median shelf anterior to the median palatal ridge. This shelf is bounded laterally by elongate grooves, which continue posteriorly along the edges of the tall, blade-like median palatal ridge.

The septomaxilla is a plate-like bone confined to the naris. The naris in *Dicynodon angielczyki* is relatively large, and in addition to the actual opening into the nasal cavity incorporates a wide, rounded embayment of the lateral snout surface composed of premaxilla, septomaxilla, and maxilla (Figs. 9, 10A and 10B). The posterior edge of this embayment is bounded by a sharp, near-vertically-oriented ridge running posterodorsal to anteroventral on the maxilla. The ventral edge of the embayment features a shallow groove in the maxillary surface, immediately lateral to its contact with the septomaxilla. Ventrally, the maxilla forms a pointed caniniform process (Fig. 9) housing the tusks. The tusks of UMZC T1089 are broken off (Figs. 8C and 8D), but other specimens of *Dicynodon angielczyki* in which they are preserved (Fig. 2C) show that they were directed

anteroventrally, as in *Dicynodon lacerticeps*. The posterior face of the caniniform process is depressed, such that its margin is distinctly concave in cross-section (Figs. 8C and 8D). The bone surface of this depression is smooth, unlike the rugose lateral surface of the maxilla. The posterior process of the maxilla that extends into the zygomatic arch is relatively short in *Dicynodon angielczyki*, as it is "crowded out" by the enlarged anterior contributions of the squamosal and jugal. It has only a thin, attenuating ventral extension below the posterior half of the orbit (Fig. 9).

The nasal is a broad bone making up much of the dorsal surface of the snout (Figs. 8A and 8B). The dorsal surface of the nasals is rugose and densely foraminated. A lengthy mid-nasal suture separates the premaxilla from the frontals. No anterior frontal process is present between the nasals posteriorly (like *Dicynodon lacerticeps* but unlike many other dicynodontoids; for example, frontal processes are present in *Peramodon* and, albeit more weakly developed, in *Daptocephalus*, and nearly contact the premaxilla in *Dinanomodon* and *Vivaxosaurus*; *Kammerer, Angielczyk & Fröbisch, 2011*). Discrete nasal bosses are not present, but rather take the form of a single, raised, shelf-like area extending across much of the nasal surface in the internarial region. The edges of this shelf extend as pointed projections into the naris (Figs. 10A and 10B), giving the dorsal narial margin a "notched" appearance. Ventrally, the nasal contacts a dorsal extension of the maxilla, separating the septomaxilla and lacrimal. The lacrimal is a small bone largely restricted to the anterior orbital margin, but has a small anterior contribution between the nasal and maxilla (Fig. 9). A single, large lacrimal foramen is present on the posterior face of the bone within the orbit, behind a knob-like lacrimal process (Figs. 10A and 10B).

The surface of the prefrontal is also sculptured, but with finer ornamentation than on the nasal (Figs. 9A and 10A). No prefrontal boss is present, although the dorsal orbital margin (extending across the prefrontal, frontal, and postorbital) is uniformly weakly swollen. The frontal is a broad, roughly rectangular bone making up most of the dorsal orbital margin. No interorbital ridge is present on the mid-frontal suture; instead, there is an interorbital depression. Several large, irregular pits are present on the frontal surface in UMZC T1089 (Fig. 8A). These appear to be natural features of the bone (although they may have been exaggerated by acid preparation), and are visible in other specimens of *Dicynodon angielczyki* where the skull roof has not been overprepared (e.g., UMZC T979). Ventrally, the frontal curves to form the roof of the orbit, then extends ventromedially to contact the orbital plate (Fig. 9). This element, also known as the anterior plate, represents a fusion of the orbitosphenoid and mesethmoid (*Cluver, 1971*) and forms a median wall separating the orbits.

The jugal typically has limited lateral exposure in dicynodontoids, mostly forming a thin strip below the anterior half of the orbit. In *Dicynodon angielczyki*, the jugal is greatly expanded in size relative to *Dicynodon lacerticeps* and other Permian dicynodontoids, with a tall contribution forming the base of the postorbital bar and separating the postorbital bone from the squamosal (Figs. 7C and 9). Ventrally, it forms much of the medial surface of the subtemporal arch, curving anteromedially to cover the posterior face of the maxilla (Figs. 8C and 8D). In this region, the jugal, maxilla, and palatine bound a large

labial fossa, as is typical of dicynodontoids (*Angielczyk & Kurkin, 2003*; *Kammerer & Angielczyk, 2009*).

The zygomatic ramus of the squamosal is greatly expanded at its anterior end, terminating in a broadly-rounded tip that covers most of the suborbital portion of the maxilla in lateral view (Fig. 9). Its greatest expansion is immediately posterior to the postorbital bar (Figs. 2C, 7C and 9), resulting in the temporal arch being sharply bowed in this area (Figs. 2A, 7A, 8A and 10A). The subtemporal zygoma is horizontally-oriented, as in *Dicynodon lacerticeps*, in contrast to the condition in *Daptocephalus* where it is more vertical. Also as in *Dicynodon lacerticeps*, the posterior contact between the zygomatic and quadrate squamosal rami forms an acute angle (Figs. 2C, 7C and 9) rather than a broadly-rounded arc (as in *Daptocephalus*). The anterolateral surface of the quadrate squamosal ramus is strongly depressed, producing a fossa for attachment of the M. adductor mandibulae externus lateralis (*Angielczyk, Hancox & Nabavizadeh, 2018*). Posteriorly, the squamosal has a large contribution to the lateral edge of the occipital plate (Figs. 10C and 10D). The dorsal and lateral edges of the squamosal are attenuate occipitally, and extend somewhat posterior to the main portion of the occipital plate. Medially on the occiput, the squamosal surface is depressed, particularly where it forms the lateral margin of the post-temporal fenestra.

The preparietal surrounds the anterior half of the oval pineal foramen (Figs. 2A, 7A, 8A and 8B). Its posterior portion is bounded laterally by thin anterior processes of the parietals. In all specimens the preparietal expands in transverse width anteriorly, and in specimens with well-exposed sutures (e.g., UMZC T1089; Fig. 8A) the anterior margin is ragged with three distinct tips. The preparietal surface is depressed, and this depression is not contiguous with the parallel depressions on the posterior frontal processes and postorbitals.

The postorbital contribution to the postorbital bar is gently curved and expands in width ventrally (Figs. 8A and 8B), where it overlies the jugal. The anterodorsal margin of the postorbital is somewhat rugose and forms part of the generally swollen dorsal rim of the orbit. The raised anterior edge of the postorbital contribution to the skull roof bounds a deep, falciform depression that would have served as an attachment site for jaw adductor musculature. A distinct postfrontal is not present in *Dicynodon angielczyki*, but it is possible that this anterior portion of the postorbital incorporates the postfrontal, as a postfrontal is present in this region in *Dicynodon lacerticeps* (*Cluver & King, 1983*; *Kammerer, Angielczyk & Fröbisch, 2011*) and fusion between the postfrontal and postorbital over the course of ontogeny occurs in other dicynodonts (*Kammerer & Smith, 2017*; *Angielczyk, Hancox & Nabavizadeh, 2018*). The postorbital contribution to the intertemporal bar is relatively broad and more horizontally-oriented than vertical. It makes up the entire medial margin of the temporal fenestra and terminates in a curved process along the posterior edge of the temporal fenestra, overlying the occiput. At its medial border the postorbital forms a thin crest around the parietal. The parietals are barely visible dorsally, with only very narrow exposure between the postorbitals and short anterior processes around the preparietal. Laterally, the parietal is exposed below the postorbitals as
part of the dorsal portion of the braincase, and is visible where it contacts the ascending process of the epipterygoid (Fig. 9).

The anteriormost visible portion of the vomer is immediately behind the median palatal ridge of the premaxilla, and continues as a similar blade-like structure posteriorly (Figs. 8C and 8D). Because the palate has been fully acid-prepared in UMZC T1089, the dorsal portion of the vomer is also visible in ventral view, a rarity among dicynodont specimens. It forms paired, vaulted laminae curving ventrolaterally to contact the palatines laterally and pterygoids posterolaterally. Medially, it surrounds an elongate, "teardrop"-shaped interpterygoid vacuity (narrow end anterior), bearing thin ridges along the margin of the vacuity.

A distinct ectopterygoid is not clearly present in any of the known specimens of *Dicynodon angielczyki*, but sutural edges of what may be the ectopterygoid are visible in UMZC T982 and UMZC T1123. The ectopterygoid is definitely absent as a separate ossification (either not ossifying or fused with one the surrounding bones, probably the maxilla) in many Triassic dicynodonts, such as *Lystrosaurus* and some kannemeyeriiforms (*Cluver, 1971*; *Maisch, 2002*; *Angielczyk, Hancox & Nabavizadeh, 2018*), but is usually present in Permian, "*Dicynodon*"-grade dicynodontoids (*Kammerer, Angielczyk & Fröbisch, 2011*; there are exceptions, however—see *Angielczyk & Kurkin, 2003*). Here, a probable outline for the ectopterygoid is shown (Fig. 8D) based on the morphology in the aforementioned specimens, but this is tentative.

The palatine of *Dicynodon angielczyki* is typical for dicynodontoids, with a raised, rugose palatine pad anteriorly and a smooth, laminar section forming part of the lateral wall of the choana posteriorly (Figs. 8C and 8D). A small, rounded lateral palatal foramen is present between the maxilla, (possibly) ectopterygoid, and palatine, at around the midpoint of the latter. The pterygoids form a roughly X-shaped unit composed of anterior and posterior (or quadrate) rami united by a median pterygoid plate, as is the case in all dicynodonts (*King, 1988*). The anterior pterygoid rami are bowed laterally, surrounding a broad choana. Narrow ridges are present on the posterior halves of the anterior pterygoid rami, which unite at the median pterygoid plate to form a well-developed crista oesophagea. The median pterygoid plate is strongly constricted relative to the anterior and posterior rami, as is usual for *Dicynodon* (Figs. 3A, 3C, 8C and 8D) but not *Daptocephalus* (Figs. 3B, 3D and 3E). The posterior or quadrate rami are thin, ribbon-like structures extending from the median pterygoid plate toward the quadrates at a 30°–45° angle relative to the long axis of the skull. In all three specimens where these fragile structures are preserved, the posterior pterygoid rami are slightly twisted through their length (Figs. 3A, 8C and 11B). Twisting of the quadrate ramus of the pterygoid is naturally present in some therapsid taxa (e.g., gorgonopsians; *Kammerer, 2016*), but can probably be attributed to taphonomic distortion in *Dicynodon angielczyki*—this ramus is usually straight in dicynodonts (e.g., Fig. 3C) and its shape is asymmetrical in all the studied specimens of *Dicynodon angielczyki* (note that the left posterior ramus in UMZC T1123 is straight for most of its length then broken at tip, whereas twisting in the right ramus appears to be the result of the anterior face of the ramus being displaced ventrally through crushing; Fig. 11B). Dorsally, the pterygoid bears a median, laminar cultriform process extending

anteriorly and terminating below the orbits, where it is underlain by the vomer. Resting on top of the pterygoids posteriorly are paired epipterygoids, which consist of an anteroposteriorly elongate footplate and a thin ascending process, which expands dorsally where it contacts the parietal (Fig. 9).

A clear sutural boundary between the pterygoid and parabasisphenoid is not visible in any of the studied specimens, but the anterior extent of the latter can be recognized by the presence of paired, ventrally-directed internal carotid canals (Figs. 8C and 8D). Posterior to these openings, the ventral surface of the parabasisphenoid bears paired ridges (with a marked depression between them) that curve posterolaterally and expand to join the basal tubera. The tubera are semi-oval in shape and are angled somewhat ventrolaterally. A deep intertuberal depression is present medially between the parabasisphenoid and basioccipital. Only the anterior edges of the tubera are composed of parabasisphenoid; the rest is basioccipital. The basioccipital does not show distinct sutures with the opisthotic (Figs. 8B and 8C), exoccipitals, or supraoccipital (Figs. 10C and 10D) and it appears that they form a single fused element, also incorporating the prootic and thus representing a periotic. Fusion of some or all of these occipital and basicranial elements is common in dicynodonts (*Surkov & Benton, 2004*; *Boos et al., 2016*; *Angielczyk & Kammerer, 2017*). The contributions of the basioccipital and exoccipitals can still be discerned in the tripartite occipital condyle, as they each make up a distinct, knob-like process, but these processes are fused at the base (as can be seen due to damage to the condyle in UMZC T1089, revealing uniform bone internally; Fig. 10C). The paroccipital processes are large, wing-like structures in *Dicynodon angielczyki*, with a curved, protruding ventral edge and a broad depression of their dorsal posterior surface, below their contribution to the margin of the post-temporal fenestra. Dorsal to the post-temporal fenestra, what is presumably the supraoccipital bears prominent, dorsolateral-to-ventromedially angled ridges. The dorsal portion of the supraoccipital is weakly depressed on its posterior face, and its dorsal margin is split by a ventral process of the postparietal.

The postparietal (or interparietal) is a roughly trapezoidal, plate-like bone situated at the dorsal midpoint of the occiput (Figs. 10C and 10D). It does not make a noticeable contribution to the skull roof. It is bounded laterally by the tabulars, which are small, flat bones made up of a narrow dorsal process and a broader ventral plate.

The quadrate and quadratojugal are fused to form a large element anteroventral to the quadrate ramus of the squamosal (Fig. 9) and exhibit the usual morphology for dicynodonts (*King, 1988*). The quadratojugal forms a broad but thin plate separated from the dorsal portion of the quadrate by a large quadratojugal foramen or channel. It is mostly occluded in posterior view by a ventral process of the squamosal (Fig. 10D). The articular surface of the quadrate ventrally is made up of lateral and medial condyles of roughly equal size separated by a trochlea (Figs. 8C and 8D). The quadrate is angled such that the medial condyle is situated somewhat posterior to the lateral one.

The mandible (based on UMZC T1123) is edentulous and primarily composed of a large, robust dentary (Fig. 12). As usual for dicynodonts, the dentary is a single fused structure; separation of the left mandibular ramus in this specimen was managed by

cutting it through the symphysis, not through natural disarticulation of the two hemimandibles. The anterior and lateral surface of the dentary is rugose; this is at least partially the result of acid preparation but some of the rugosity on the anterior and dorsal edges of the symphysis appears natural and probably corresponds to coverage by the keratinous beak. Anterodorsally the dentary extends to a curved, pointed tip terminating well above the dorsal edge of the rest of the mandible. The border between the anterior and lateral faces of the dentary are weakly demarcated, without a sharp ridge between them. Posterolaterally, the dentary terminates in two processes above and below the mandibular fenestra. Below the mandibular fenestra, the posterior process of the dentary is triangular and fits into a deep facet on the angular. Above the fenestra, the dentary has a longer posterior process, which attenuates somewhat posteriorly but does not come to a discrete point. Rather, the posterior edge of the dentary in this region remains tall where it overlies the surangular. The posterior margin of this process has a distinct concavity, as figured by *Cluver & King (1983)* for South African *Dicynodon*. A lateral dentary shelf is present at the anterodorsal edge of the mandibular fenestra. It is weakly developed and thin, as in many other Permian bidentalians, but unlike *Aulacephalodon* where it extends anteriorly to join an enlarged, rounded boss (*Kammerer, Angielczyk & Fröbisch, 2011*).

The mandibular fenestra is elongate and bounded dorsally by the dentary and surangular and ventrally by the angular. The splenial, like the dentary, is a single fused element. It forms a large portion of the ventral margin of the symphysis (as is typical of dicynodontoids; see *Kammerer, 2018*) and continues posteriorly along the medial face of the jaw, terminating in an attenuate process overlying the angular (Fig. 12E). A thin process of the angular extends forwards between the splenial and dentary to contribute to the symphysis. Posteriorly, the angular consists of a flat, ribbon-like element covered by dentary laterally and splenial medially. It is well-exposed below the mandibular fenestra and bears a posteroventrally-directed reflected lamina beneath the posterior edge of the fenestra. The reflected lamina is somewhat damaged but appears typical for dicynodontoids: it is a free-standing structure (i.e., the posterior edge is not bound to the main jaw ramus) and bears one major ridge surrounded by two surficial concavities (with weaker ridges along the anteroventral and posterodorsal edges of the lamina). A flat portion of angular forms most of the lateral surface of the jaw posterior to the reflected lamina.

The surangular and prearticular are clearly separate medially (Fig. 12E), but the distinction between these bones and the articular is unclear and at least partial fusion between these three elements is likely. Laterally, the surangular is exposed as a narrow strip above the angular along the dorsal edge of the post-dentary jaw ramus. Due to taphonomic distortion, it is somewhat displaced in UMZC T1123 and would originally have had a greater degree of lateral exposure. The prearticular is also damaged, with its anterior tip broken off. When complete it would have extended anteriorly to overlie the splenial. Medially, both the surangular and prearticular are thin, ribbon-like elements bracing the angular wall. The articular morphology is similar to that of other dicynodontoids, consisting of lateral and medial condyles around a median trochlea, where it would

articulate with the quadrate (Fig. 12C). No retroarticular process is evident, but this is probably attributable to damage, as this structure is present in other *Dicynodon* (*Cluver & King, 1983*).

## PHYLOGENETIC ANALYSIS

*Dicynodon huenei* has been a problematic taxon in recent analyses of dicynodont phylogeny. In part, this is due to the general instability of the Permian dicynodontoid portion of the tree (see discussion in *Angielczyk & Kammerer, 2017*), but here, its problematic status is also recognized as the result of chimaerical codings, including data from what are probably three distinct species (*Daptocephalus huenei*, *Dicynodon angielczyki*, and so-called "*D. huenei*"/"*Dicynodon trigonocephalus*" from Zambia). *Kammerer, Angielczyk & Fröbisch's (2011)* were the first to include *D. huenei* in a cladistic analysis of anomodont phylogeny, and recovered it as the sister-taxon of *Dicynodon lacerticeps* in their primary tree. However, support for this relationship was extremely low, and in variant analyses the genus *Dicynodon* sensu Kammerer et al. (i.e., containing *Dicynodon lacerticeps* and *D. huenei*) was not found to be monophyletic. *Kammerer, Angielczyk & Fröbisch (2011)* discrete state codings for *D. huenei* were based on cranial data mostly from UMZC T1089, mandibular data mostly from UMZC T1123, and postcranial data from the Zambian specimen NHMUK PV R37005 (=TSK 14), and continuous codings were based on GPIT/RE/7175 (=K110), GPIT/RE/7177 (=K101), GPIT/RE/9316 (=K2), NMT RB43, NHMUK PV R37005, NHMUK PV R37374 (=TSK 37), UMZC T979, UMZC T982, UMZC T987, UMZC T1089, and UMZC T1123.

In the current analysis, the previous "*Dicynodon huenei*" operational taxonomic unit (OTU) has been deleted and replaced with separate OTUs for *Dicynodon angielczyki* (coded based on GPIT/RE/7175, GPIT/RE/7177, UMZC T979, UMZC T982, UMZC T1089, and UMZC T1123) and *Daptocephalus huenei* (coded based on GPIT/RE/9316, GPIT/RE/9317, GPIT/RE/9641, NMT RB43, and SAM-PK-10630). These OTUs were added (see Supplemental Information) to the most recent iteration of the anomodont character matrix originally published by *Kammerer, Angielczyk & Fröbisch (2011)*, namely that of *Kammerer et al. (2019)*. The emydopoid *Thliptosaurus imperforatus* (*Kammerer, 2019*) was also added to that analysis, but the recently-described Laotian dicynodontoids *Counillonia superoculis* and *Repelinosaurus robustus* (*Olivier et al., 2019*) were not included, as their holotypes were not available for study during the production of this paper. The data were analyzed using parsimony in TNT v1.5 (*Goloboff, Farris & Nixon, 2008*) using New Technology search parameters (tree drifting, parsimony ratchet, and tree fusing), starting at level 65 and forced to find the shortest tree at least 20 times. Discrete-state characters 58, 61, 79, 140, 150, 151, and 166 were treated as ordered. Symmetric resampling values were calculated based on 10,000 replicates.

Two most parsimonious trees of length 1,158.54 were recovered (consistency index = 0.238, retention index = 0.720), differing only in the positions of genera within Placeriinae (consensus topology for bidentalian dicynodonts shown in Fig. 13). The overall tree topology is very similar to that of *Kammerer (2019)* and *Kammerer et al. (2019)*, and the relationships of non-bidentalian anomodonts in the most parsimonious trees are identical

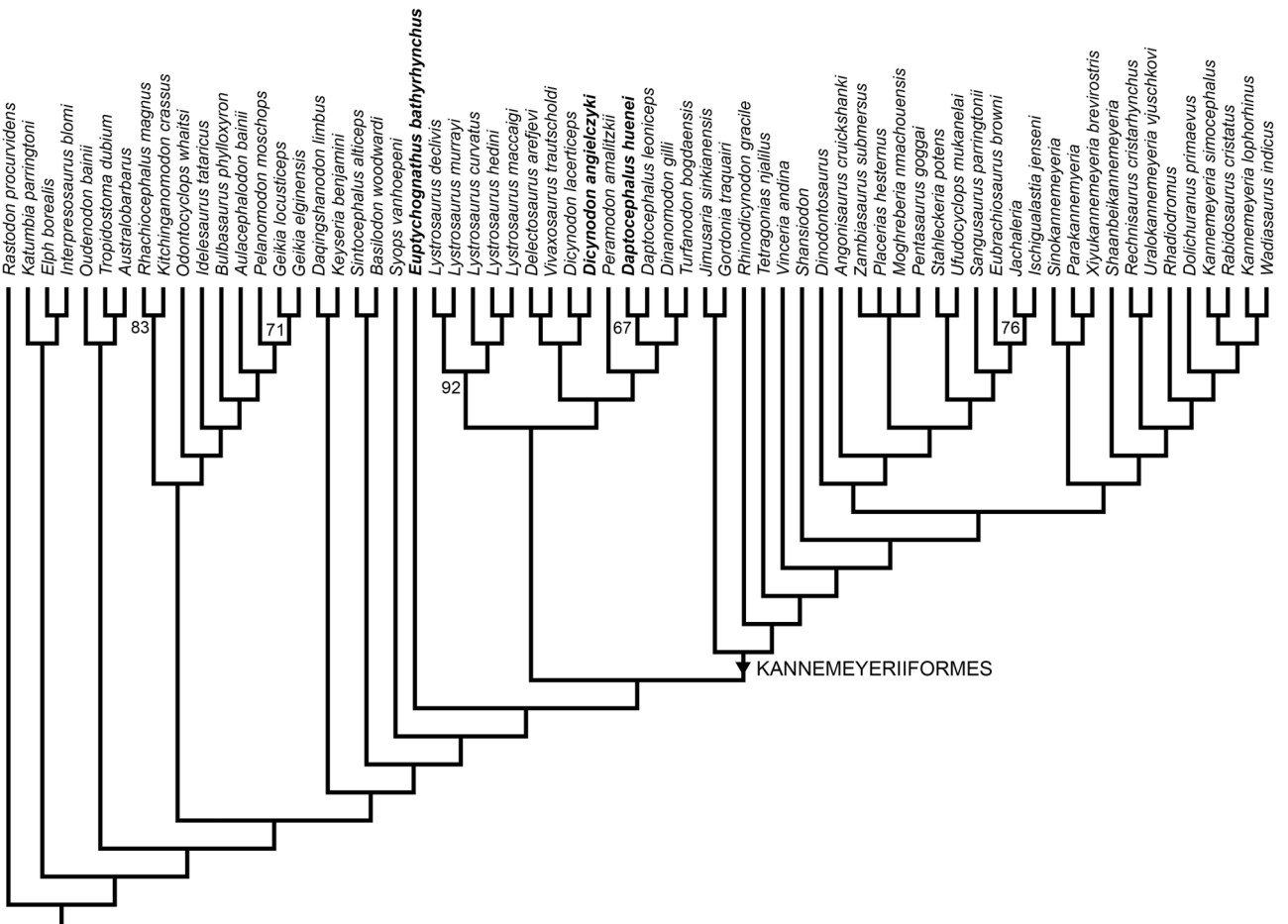

**Figure 13 Phylogeny of Bidentalia.** Usili Formation dicynodontoids in bold. Numbers at nodes represent symmetric resampling values >50.

to those of *Kammerer (2019)*. Relationships among bidentalians remain, with few exceptions, poorly-supported and labile. Unlike the analysis of *Kammerer et al. (2019)*, but as in *Kammerer (2019*; see also *Boos et al., 2016*; *Angielczyk & Kammerer, 2017*; *Olroyd, Sidor & Angielczyk, 2018*), Cryptodontia is not recovered as monophyletic in its traditional sense (i.e., rhachiocephalids + geikiids are more closely related to "*Dicynodon*"-grade dicynodontoids than oudenodontids). However, despite continued uncertainty surrounding the intergeneric relationships of bidentalians, the current analysis does recover *Dicynodon* (*Dicynodon lacerticeps* + *Dicynodon angielczyki*) and *Daptocephalus* (*Daptocephalus leoniceps* + *Daptocephalus huenei*) as monophyletic, with *Daptocephalus* being one of the few strongly supported clades in the resampling analysis (Fig. 13). The *Dicynodon* clade is supported by two characters (continuous character 14, shape of mandibular fenestra, and discrete state character 51, postorbital contribution to intertemporal bar relatively flat) and the *Daptocephalus* clade is supported by four characters (continuous characters 8, median pterygoid plate width, and 11, relative area of internal nares, and discrete state characters 92, ventral surface of median pterygoid plate

smooth and flat, and 111, absence of central circular depression on occipital condyle) (refer to Supplementary Information for lists of all character states).

## DISCUSSION

The discovery that multiple taxa of large dicynodontoids are present in the Usili Formation should not be surprising—this clade is common in the late Permian, and sympatric dicynodontoid taxa are known within more poorly-sampled, probably coeval Laurasian basins (*Li, Wu & Zhang, 2008*; *Kurkin, 2012*; *Olivier et al., 2019*) in addition to the heavily sampled Karoo Basin of South Africa (*Kammerer, Angielczyk & Fröbisch, 2011*). What is notable about the Usili species, however, is their close similarity to (and recovery as sister-taxa of) particular Karoo species (*Dicynodon lacerticeps* and *Daptocephalus leoniceps*), to the extent that they are here considered congeneric. Although genera are an arbitrary taxonomic construct, in the current case the differences between the Tanzanian and South African species are so few (especially in the case of *Daptocephalus huenei* vs. *Daptocephalus leoniceps*) that generic separation seems unwarranted.

The present work on "*Dicynodon huenei*" has benefited from a detailed view of dicynodont variation only possible after decades of taxonomic revision of the group (see reviews in *Kammerer & Angielczyk, 2009*; *Kammerer, Angielczyk & Fröbisch, 2011*). Research on the expansive sample of fossils from the Karoo Basin in particular has provided substantial insight into what is most parsimoniously interpreted as intraspecific (mostly ontogenetic, sexual, and taphonomic) variation in dicynodont species, which was often historically interpreted as representing distinct species (*Broom, 1932*). Given the unwieldy and egregiously oversplit nature of historical taxonomic schemes for dicynodonts (recognized as such even at the time; see review in *Kammerer, Angielczyk & Fröbisch, 2011*), latter-day revisions of dicynodont taxonomy have understandably focused on synonymizing the many nominal taxa. However, another result of the more nuanced view of dicynodont taxonomy currently available is the ability to tease apart instances of overlumping: cases where previously-synonymized taxa are shown to be distinct (*Kammerer, Angielczyk & Fröbisch, 2015*) or where some specimens previously referred to common taxa represent unrecognized species (*Kammerer, Bandyopadhyay & Ray, 2016*; *Kammerer & Smith, 2017*). Such cases are likely to increase in number as previously poorly-sampled basins become better known, and taxa previously known from limited material once difficult to distinguish from Karoo species become more robustly diagnosed. For example, *Keyser (1975)* recognized only a single valid rhachiocephalid species, *Rhachiocephalus magnus*, including material from the Usili Formation, but more recent study (*Maisch, 2005*) has shown that the Tanzanian specimens are specifically distinct from South African *Rhachiocephalus*. *Kammerer, Angielczyk & Fröbisch (2011)* could not find consistent differences separating Zambian specimens of *Oudenodon* (previously known as *O. luangwanensis*) and the South African type species *O. bainii*, and considered only *O. bainii* to be valid. However, the type materials of Zambian *Oudenodon* they examined were generally poor (the holotype of *O. luangwanensis*, SAM-PK-11310, for example, is almost entirely unprepared, with only the skull roof exposed), and would not necessarily have shown the "species-level" variation recognized

for example, *Daptocephalus* here. *Angielczyk et al. (2014)* figured additional, complete and well-prepared Zambian *Oudenodon* specimens, and although they also referred these specimens to *O. bainii* (following *Kammerer, Angielczyk & Fröbisch, 2011*), the new specimens do show some consistent proportional differences from typical South African *O. bainii*. Even more recently-discovered, well-preserved Zambian *Oudenodon* specimens are now known (K Angielczyk, 2019, personal communications). Detailed study of these specimens is needed to determine whether they form a discrete morphotype from Karoo specimens, but preliminary information is suggestive of their distinction. As a final example, *Angielczyk (2019)* recently described the first specimen of the rare dicynodont *Digalodon* (previously known only from South Africa) from the Luangwa Basin of Zambia. Although clearly referable to *Digalodon*, the Zambian specimen (NHCC LB830) differs from South African skulls in several notable regards (non-diverging anterior median palatal ridges, rounded postcaniniform keel, horizontal zygoma, continuous rim of the pineal foramen, proportionally broader intertemporal bar). *Angielczyk (2019)* considered this specimen taxonomically uncertain and classified it as *Digalodon* cf. *D. rubidgei*, a reasonable approach given its singleton nature and our poor knowledge of variation in South African *Digalodon* (known from a few, mostly poorly-preserved and prepared specimens; *Kammerer, Angielczyk & Fröbisch, 2015*). However, the unique features of NHCC LB830 are not known to vary intraspecifically in other emydopoids, and it is likely to represent a distinct species.

The emerging pattern of "low-level" or "species-level" endemism among dicynodonts in the African late Permian basins is interesting in the context of recent proposals concerning changing biogeographic patterns across the Permo–Triassic boundary. Based on network analyses of Permo–Triassic vertebrate assemblages, *Sidor et al. (2013)* argued that there is a sharp increase in provincialism among tetrapods between the Permian and Middle Triassic. It is true that in the late Permian, dicynodont genera such as *Pristerodon*, *Oudenodon*, *Dicynodontoides*, and *Endothiodon* (cited by *Sidor et al. (2013)* as cosmopolitan taxa) have a broad, interbasinal distribution, and that Middle Triassic faunas show greater taxonomic/phylogenetic separation (at least between the African basins, though some evidence suggests greater similarity between Zambian/Tanzanian and coeval South American assemblages; see *Peecook et al., 2018*). The recognition that a number (potentially many) of these wide-ranging dicynodont genera contain multiple, locally endemic species adds a new wrinkle to this proposal, however. I would suggest that rather than a simple transition from cosmopolitan Permian tetrapod faunas to provincialized Triassic ones following the Permo–Triassic mass extinction, there is a shift between "weakly provincialized" faunas (substantial phylogenetic propinquity between basins, but frequent species-level endemism in each) in the late Permian, to true cosmopolitanism associated with the spread of "disaster taxa" in the wake of the Permo–Triassic extinction (as in individual species like *Lystrosaurus murrayi* with well-supported circum-Gondwanan distributions; *Colbert, 1974*; *Ray, 2005*), and finally "strongly provincialized" faunas in the Middle Triassic (distinct at higher taxonomic levels; probably, as *Sidor et al. (2013)* argued, resulting from heterogeneous re-occupation of empty ecospace during ecosystem recovery). Continued research on late Permian faunas,

particularly from basins outside of the well-sampled Karoo, is needed to test this proposal. Additional paleoecological data is also needed—given the apparent importance of local climate in driving Triassic tetrapod distributions (*Whiteside et al., 2011*), it needs to be determined whether the heterogeneous repopulation of faunas in the Middle Triassic is stochastic, or whether taxon composition was driven by local environments.

## CONCLUSIONS

*Dicynodon huenei*, a supposed taxon of "*Dicynodon*"-grade dicynodontoid from upper Permian strata of Tanzania and Zambia, is here recognized as being made up of several distinct dicynodont species. The Tanzanian specimens of "*D. huenei*" constitute two species, here reclassified as *Daptocephalus huenei* comb. nov. and *Dicynodon angielczyki* sp. nov. Permian dicynodontoids have proven to be one of the most troublesome regions of dicynodont phylogeny, exhibiting substantial instability between recent phylogenetic analyses. Separation of "*Dicynodon huenei*" into multiple OTUs resolves some problems in recent analyses (i.e., occasional polyphyly of the genus *Dicynodon*), but support for the current topology remains low. Re-evaluation and expansion of the character data for dicynodontoids is required for better resolution in this part of the tree.

## INSTITUTIONAL ABBREVIATIONS

| | |
|---|---|
| **BP** | Evolutionary Studies Institute, University of the Witwatersrand, Johannesburg, South Africa |
| **GPIT** | Paläontologische Sammlung, Eberhard-Karls-Universität Tübingen, Germany |
| **MB** | Museum für Naturkunde, Berlin, Germany |
| **NHCC** | National Heritage Conservation Commission, Lusaka, Zambia |
| **NHMUK** | The Natural History Museum, London, UK |
| **NMT** | National Museum of Tanzania, Dar es Salaam, Tanzania |
| **PIN** | Paleontological Institute of the Russian Academy of Sciences, Moscow, Russia |
| **RC** | Rubidge Collection, Wellwood, Graaff-Reinet, South Africa |
| **SAM** | Iziko: The South African Museum, Cape Town, South Africa |
| **TSK** | Former collections of Prof. Tom Kemp (Oxford, UK), now NHMUK specimens |
| **UMZC** | University Museum of Zoology, Cambridge, UK. |

## ACKNOWLEDGEMENTS

Thanks to Ken Angielczyk for invaluable assistance during this project, including permitting me to examine undescribed dicynodont specimens collected in the recent TZAM expeditions. Thanks also to Christian Sidor and Roger Smith for various discussions on the fauna and stratigraphy of the Usili Formation. I thank Paul Barrett, Sifelani Jirah, Tom Kemp, Mathew Lowe, Robert and Marion Rubidge, Zaituna Skosan, and Ingmar Werneburg for access to specimens in their care, and Michael Day for catalog information on recently-accessioned NHMUK specimens. Reviews by Jun Liu, Savannah Olroyd, and Brandon Peecook improved the manuscript, as did the efforts of editor Andrew Farke.

### Funding

This work was supported by a grant from the Deutsche Forschungsgemeinschaft (KA 4133-1/1) and from the SYNTHESYS Project (http://www.synthesys.info/), which is financed by European Community Research Infrastructure Action under the FP7 Integrating Activities Programme. There was no additional external funding received for this study. The funders had no role in study design, data collection and analysis, decision to publish, or preparation of the manuscript.

### Grant Disclosures

The following grant information was disclosed by the authors:
Deutsche Forschungsgemeinschaft: KA 4133-1/1.
SYNTHESYS Project.
European Community Research Infrastructure Action under the FP7 Integrating Activities Programme.

### Competing Interests

The author declares that they have no competing interests.

### Author Contributions

- Christian F. Kammerer conceived and designed the experiments, performed the experiments, analyzed the data, prepared figures and/or tables, authored or reviewed drafts of the paper, approved the final draft.

### Data Availability

The character matrix for the phylogenetic analysis, character list, and continuous character data are available as Supplemental Files.

The specimens described herein are housed in the publicly accessible fossil collections of the following institutions:

Eberhard-Karls-Universität Tübingen (Germany), Iziko (Cape Town, South Africa), and the University Museum of Zoology (Cambridge, UK).

Accession numbers for these specimens are: GPIT/RE/7175, 7177, 9316, 9317, 9641, SAM-PK-10630, UMZC T799, T979, T982, T983, T1089, T1123, T1126, T1282.

### New Species Registration

The following information was supplied regarding the registration of a newly described species:

Publication LSID: urn:lsid:zoobank.org:pub:714AFA18-EB4D-4A35-B948-B4E7CB046A77

*Dicynodon angielczyki* LSID: urn:lsid:zoobank.org:act:5C7C1D87-8F39-4E7C-8E23-7015FBD545A0

## Supplemental Information

Supplemental information for this article can be found online at http://dx.doi.org/10.7717/peerj.7420#supplemental-information.

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
