# Peer review of "Revision of the Tanzanian dicynodont Dicynodon huenei (Therapsida: Anomodontia) from the Permian Usili Formation"

_PeerJ, doi:10.7717/peerj.7420_

## Round 0.1 · original submission · Minor Revisions

- Please include a full character list, not just character codings, in the supplemental information. I think it is important for papers to be self-sufficient in explaining phylogenetic analyses, especially if character lists are behind paywalls or are in press. (Reviewer 2 also mentions this)

- If possible, please include some basic measurements of the specimens studied here (e.g., cranial width, length, etc.; or relevant individual bones). I recognize this may not be feasible given the broad geographic dispersal of these fossils, so categorize this suggestion as "strongly recommended but not mandatory."

·

Basic reporting

Kammerer examines the phylogenetically problematic dicynodontoid species from the upper Permian of Tanzania, 'Dicynodon huenei'.

After an authoritative review, he walks us through the referral of Tanzanian specimens of 'Dicynodon huenei' to Daptocephalus huenei (new combination) and erects Dicynodon angielczyki (new species), based on consistent morphological differences that are well figured. Kammerer also clearly explains consistent differences between Tanzanian Daptocephalus and Dicynodon and their congeners in South Africa. The phylogenetic analysis supports the morphological interpretations, and the assignments fit into a developing picture of species level endemism among dicynodonts across upper Permian Pangea.

A few suggestions/edits:
-Specimen numbers for comparative taxa on lines 363-364 in the description (Peramodon, Vivaxosaurus, Dinanomodon) would be beneficial (with corresponding institutional abbreviations added). In-text comparison with Karoo Daptocephalus and Dicynodon do not need numbers necessarily as that information is included in figure legends.
-Figure 8: "pf, pineal foramen" is in the legend, but not labelled on the figure; "mpp" is labelled on the figure, but not listed among the anatomical abbreviations.
-Figure 11: The legend refers to "Figure Y" when it should say "Figure 12".

Experimental design

The OTUs were included in extremely modern phylogenetic analyses.

Validity of the findings

No comment

·

Basic reporting

The paper is written clearly and logically. There is a wealth of relevant references that will be extremely helpful for readers. The figures also look great and are organized in a way that facilitates comparisons between specimens. I have a few changes to suggest that may help with clarity:
Line 65: As tedious as it is, since each formation is a proper name you should include "Formation" captialized after each one. "Usili Formation and Madumabisa Mudstone Formation". Don't worry, we all hate it, too.
Line 128: Is there any reason to not call out a figure here for UMZC T1089 to show the thickened zygomatic arch in this specimen?
Line 129: The thickened zygomatic arch doesn't look obvious to me in Figure 3A. I'm guessing that's because the matrix is covering the arch a bit in this view, so maybe choose another view to illustrate this or at least mention that the thickened arch is partially obscured by matrix in this figure.
Line 130: Change to “…present in SAM-PK-10630, and Angielczyk…”
Line 138: Add a period after “(Kammerer, pers. obs.)”
Lines 151-161: I’m not sure the numbering here is very helpful. 1-4 would work just fine as individual sentences, and the rest could just be separated by semicolons. The numbering makes it a little messy, and it’s even a bit confusing when items that aren’t complete sentences have the first letter capitalized.
Lines 270-271: “All referred material is from the Ruhuhu Basin of southwestern Tanzania.” Maybe move this line to the start of the paragraph so readers don't encounter the name Kingori and wonder where/what that is.
Lines 376-377: Description of the frontal shape is kinda messy/unclear. Change to "The frontal is depressed in the interorbital region."
Lines 396-399: This description is confusing. I'm not sure what you mean by height here, and how that is deceptive.
Lines 416-418: I am unclear on where this fossa is exactly. In which direction is it bounded by the postorbital? Which face of the intertemporal bar is this? Are you referring to the fossa on the underside of the intertemporal bar?
Line 446: Change to "...but this is tentative". Otherwise the wording could be confusing to non-native English speakers
Line 485: Change to: "fused at the base"
Lines 503-587: This sections jumps out of nowhere, at least the way it's introduced. I would suggest either putting this paragraph at the beginning of the description (near where you state which specimens are being mostly used for the cranial description) or putting in subheadings for the cranium and the mandible so the reader knows you're going to do another specimen introduction here. Otherwise we're suddenly reading about gorgonopsians for half a paragraph before being told why.
Figure 6 caption: “Major facial bones colored to show arrange in various specimens…” Change “arrange” to “arrangement”
Figure 11A: Unless you were to label what all these other bones are in the block, it seems to me that the only function of Figure 11A is to give context for the otherwise odd-looking 11B. To make it more useful for this purpose, I would suggest outlining where in Figure 11A the stuff in Figure 11B comes from (so, the dicynodont skull and the gorgon bones that have been grayed out.
Figure 13: Are these all the resampling values you got? If so, fine, but if not, I'm unclear on why you're displaying only the values here. It seems to me like you'd at least want to display the value for D. lacerticeps+D. angielczyki. I'd prefer to see all of the values in this figure because so many bidentalian groups have been volatile in the various iterations of this analysis (especially Cryptodontia, which you even discuss in this paper), that it could be helpful for readers to be able to easily compare node support between analyses.

Experimental design

The author does a great job of providing the historical context for this revision. He provides a thorough description and sensible taxonomic changes.
My main concern is with the lack of information on the phylogenetic analysis. I understand that this is the next installation in a series of analyses with one database, but I’d still like to see the list of characters used here as a supplemental file. If the characters haven’t changed from a previous version, then maybe just cite the paper with the version you used so readers can get this information somewhere. You do cite a paper of yours that’s in press, but even if that is published before this one, there is a level of uncertainty with citing an in-press paper as the source for your characters. It may not be clear to the audience which paper this is referring to, so I think it's best to just upload the character data here, too. I don't see any harm in doing so. Similarly, I think you could include more information about how you conducted the analysis itself. Are there any other search parameters that you personalized? Which search algorithms did you use, etc.? Were any characters treated as ordered? If these things are also identical to a previous paper, please state which one. If it's from the in-press paper, I would again urge you to go ahead and list them here, too, to make it easier to replicate your work precisely. For example, I was unable to replicate the tree length you got in your results when I ran the data you provided. I even tried running it with parameters you mentioned in your emydopoid paper, and it still didn’t give me the tree length you got here (the topology was basically the same, though).

Validity of the findings

Overall, this is a welcome contribution to dicynodont taxonomy. I also think the discussion about the nuances of cosmopolitanism versus endemism across the P-Tr boundary is important. I only have a few comments:
Line 253: I don’t think the “tip of varying length” belongs here in the diagnosis. You can’t identify a species based on this. I would just include the part about it lacking the hooked shape of Dinanomodon.
Lines 276-278: Based on the figure, I don't see how the lacrimal is enlarged. It seems to have a more rounded shape, but it doesn't seem to be all that different in area from that of D. leoniceps. If what you're describing is more of a shape difference, please describe it as such to make the diagnosis more precise. If there really is a size difference, maybe include another specimen or a different view that shows this more clearly in Figure 6. Additionally, based on the data given, I don't have a good sense for how important of a difference this is. I think this could be helped by including more information about how consistent this difference is. How many specimens of each species show this difference? Is there maybe an additional specimen of D. huenei that you could add into Figure 6 to further illustrate that this is a consistent difference? This would make it more convincing that these should be considered two different species based on the lacrimal proportions.

Additional comments

Overall, I think this is a great paper with thorough background information and interesting implications. I think both the taxonomic revision and the points brought up in the discussion are timely.

·

Basic reporting

Please check all for: GPIT/RE/9316 (=K6), GPIT/RE/9136(=K2), GPIT/RE/9641 (=K2),
9136 should be 9316?

L272: you should specific lacrimal taller (but shorter) (not larger) in D. huenei

Experimental design

Why not use TNT 1.5?

I tried your matrix, the shortest tree is 1153.408, there are 2 MPTs, but the consensus tree is same.

Validity of the findings

no comments

---

## Round 0.2 · accepted · Accept

Thank you for your close attention to the reviewers' comments. The paper is ready to move forward in the process.